# The late-onset Alzheimer's disease risk factor RHBDF2 is a modifier of microglial TREM2 proteolysis

Georg Jocher[1,2] , Gozde Ozcelik[1,2], Stephan A Müller[1,2], Hung-En Hsia[1,2], Miranda Lastra Osua[1,2], Laura I Hofmann[1,2], Marlene Aßfalg[1,2], Lina Dinkel[1], Xiao Feng[1,2], Kai Schlepckow[1], Michael Willem[3], Christian Haass[1,3,7], Sabina Tahirovic[1], Carl P Blobel[4,5,6] , Stefan F Lichtenthaler[1,2,7]

The cell surface receptor TREM2 is a key genetic risk factor and drug target in Alzheimer's disease (AD). In the brain, TREM2 is expressed in microglia, where it undergoes proteolytic cleavage, linked to AD risk, but the responsible protease in microglia is still unknown. Another microglial-expressed AD risk factor is catalytically inactive rhomboid 2 (iRhom2, RHBDF2), which binds to and acts as a non-catalytic subunit of the metalloprotease ADAM17. A potential role in TREM2 proteolysis is not yet known. Using microglial-like BV2 cells, bone marrow–derived macrophages, and primary murine microglia, we identify iRhom2 as a modifier of ADAM17-mediated TREM2 shedding. Loss of iRhom2 increased TREM2 in cell lysates and at the cell surface and enhanced TREM2 signaling and microglial phagocytosis of the amyloid $\beta$-peptide (A$\beta$). This study establishes ADAM17 as a physiological TREM2 protease in microglia and suggests iRhom2 as a potential drug target for modulating TREM2 proteolysis in AD.

## Introduction

Variants of the gene "triggering receptor expressed on myeloid cells 2" (TREM2) are major genetic risk factors for late-onset Alzheimer's disease, frontotemporal lobar degeneration, and the Nasu–Hakola disease (Paloneva et al, 2002; Guerreiro et al, 2013; Guerreiro et al, 2013; Jonsson & Stefansson, 2013; Rayaprolu et al, 2013; Borroni et al, 2014; Cady et al, 2014; Cuyvers et al, 2014). TREM2 acts as a cell surface receptor and is expressed in myeloid cells, including microglia in the brain (Guerreiro et al, 2013). TREM2 binds various ligands, including lipids, lipoproteins, and the amyloid $\beta$ (A$\beta$) peptide, a key pathogenic peptide early in AD pathogenesis (Deczkowska et al, 2018a; Lewcock et al, 2020; Hou et al, 2022). Ligand binding induces cell-autonomous TREM2 signaling through the

"spleen tyrosine kinase" (SYK) and controls function and activity of microglia, including their phagocytic activity, metabolic state, survival, proliferation, migration, and cytokine release, thereby regulating microglial responses to pathological challenges (Ulland et al, 2017; Ennerfelt et al, 2022; Wang et al, 2022). TREM2 signaling is required for microglia to adopt a cell state referred to as "disease-associated microglia" (DAM), in which microglia may phagocytose A$\beta$, shield plaques, and delay tau pathology, suggesting a protective role of DAM in AD (Keren-Shaul et al, 2017; Deczkowska et al, 2018b; Kodama et al, 2020). Given the central role of TREM2 signaling in microglial activity and AD, TREM2 has become an important AD drug target. Several antibodies, which bind the extracellular domain of TREM2 and activate TREM2-mediated microglial responses, are tested as immunotherapies for the treatment of AD in clinical trials (Schlepckow et al, 2023).

TREM2 is a transmembrane protein that undergoes proteolytic ectodomain shedding (Kleinberger et al, 2014), a fundamental mechanism to control the abundance and activity of membrane proteins (Lichtenthaler et al, 2018). TREM2 shedding occurs at a membrane-proximal peptide bond within the TREM2 ectodomain (Feuerbach et al, 2017; Schlepckow et al, 2017; Thornton et al, 2017). This proteolytic cleavage converts the full-length, transmembrane form of TREM2 to the soluble TREM2 ectodomain (sTREM2), which is secreted. sTREM2 may be detected diagnostically in body fluids, such as cerebrospinal fluid, and may help to monitor disease progression, in particular early in AD pathogenesis (Piccio et al, 2008, 2016; Kleinberger et al, 2014; Suárez-Calvet et al, 2016a, 2016b; Heslegrave et al, 2016; Suarez-Calvet et al, 2019; Morenas-Rodríguez et al, 2022).

TREM2 shedding may serve two distinct (patho)physiological functions. It may terminate TREM2 signaling. This is aberrantly augmented by the TREM2 H157Y variant, located directly at the TREM2 cleavage site, which enhances TREM2 shedding and increases the risk for AD (Jiang et al, 2016; Feuerbach et al, 2017;

[1]German Center for Neurodegenerative Diseases (DZNE), Munich, Germany   [2]Neuroproteomics, School of Medicine and Health, Klinikum rechts der Isar, Technical University of Munich, Munich, Germany   [3]Biomedical Center (BMC), Division of Metabolic Biochemistry, Faculty of Medicine, Ludwig-Maximilians-Universität München, Munich, Germany   [4]Department of Medicine and Department of Biochemistry, Cellular and Molecular Biology, Weill Cornell Medicine, New York, NY, USA   [5]Arthritis and Tissue Degeneration Program, Hospital for Special Surgery, New York, NY, USA   [6]Institute for Advanced Study, Technische Universität München, Garching, Germany   [7]Munich Cluster for Systems Neurology (SyNergy), Munich, Germany

Correspondence: stefan.lichtenthaler@dzne.de

Schlepckow et al, 2017; Thornton et al, 2017). Yet, soluble TREM2 can also have protective functions, for example, when injected into AD mouse brains (Zhong et al, 2019).

Despite the important role of shedding in controlling TREM2 function, the physiological TREM2-cleaving protease in microglia is not yet identified. The proteases "a disintegrin and metalloprotease 10" (ADAM10) and ADAM17, as well as meprin $\beta$, were shown to cleave TREM2 in cell lines (Kleinberger et al, 2014; Feuerbach et al, 2017; Schlepckow et al, 2017; Thornton et al, 2017; Berner et al, 2020), but it remains unknown whether the same proteases also cleave TREM2 in primary myeloid cells and specifically in microglia. Likewise, little is known about modifiers of TREM2 shedding and their contribution to AD risk and pathogenesis. Genetic variants of the MS4A gene cluster, and of TGFBR2 and NECTIN2 are associated with increased CSF sTREM2 and reduced AD risk (Deming et al, 2019; Wang et al, 2024), but the mechanism of how the MS4A proteins control TREM2 shedding remains to be established.

Besides TREM2 and the MS4A genes, variants of numerous additional genes increase late-onset Alzheimer's disease risk and several of them are selectively expressed in microglia (Andrews et al, 2023). One of them is the gene "inactive rhomboid protease 2" (iRhom2), also known as "rhomboid family member 2" (RHBDF2). Increased methylation of CpG dinucleotides is associated with enhanced iRhom2 gene expression in subjects with AD (De Jager et al, 2014; Lunnon et al, 2014; Lardenoije et al, 2019; Li et al, 2020; Palma-Gudiel et al, 2023). Whether iRhom2, similar to the MS4A genes, may also physiologically control TREM2 shedding is unknown.

iRhom2 is a multipass transmembrane protein that forms a complex with ADAM17, one of the TREM2 proteases in cell lines (Fig 1A). iRhom2 acts as a non-proteolytic subunit of the complex and binds ADAM17 in the endoplasmic reticulum (ER) (Adrain et al, 2012; McIlwain et al, 2012). Binding of iRhom2 is essential for ER exit of the complex, for activation of ADAM17 through prodomain removal in the trans-Golgi network (TGN), and for controlling proteolytic ADAM17 activity at the cell surface (Maretzky et al, 2013; Adrain & Cavadas, 2020; Weskamp et al, 2020).

iRhom1 is a homolog of iRhom2 and has a similar function in controlling trafficking, maturation, and function of ADAM17 (Christova et al, 2013; Li et al, 2015; Tüshaus et al, 2021). Although iRhom2-deficient mice are viable and fertile, deficiency of both iRhom1 and iRhom2 induces embryonic or perinatal lethality (Christova et al, 2013; Li et al, 2015). iRhom1 and iRhom2 are mostly ubiquitously expressed; however, myeloid cells, including microglia, express only iRhom2 (Issuree et al, 2013; Li et al, 2015; Lichtenthaler et al, 2015). Consequently, iRhom2-deficient mice show a selective ADAM17 loss-of-function phenotype in myeloid cells, including a loss of cleavage and release of the pro-inflammatory ADAM17 substrate TNF from microglia. Thus, iRhom2-deficient mice are largely protected from diseases involving excessively released soluble TNF, such as sepsis, rheumatoid arthritis, and lupus erythematosus glomerulonephritis (McIlwain et al, 2012; Issuree et al, 2013; Qing et al, 2018).

Given the role of iRhom2 in regulating ADAM17 function, we considered that iRhom2 may control cleavage of microglial membrane proteins beyond TNF, such as TREM2, and, thus, control microglial activation and phagocytosis. Using the mouse microglial–like BV2 cell line, bone marrow–derived macrophages, and primary microglia from iRhom2-deficient mice, we identify

ADAM17 as a TREM2 protease in microglia, establish iRhom2 as a new modifier of TREM2 shedding, and demonstrate that loss of iRhom2 increases TREM2 cell surface abundance, resulting in enhanced TREM2 signaling and microglial phagocytosis of A$\beta$.

# Results

## Loss of iRhom2 blocks ADAM17 activity in microglia

To study iRhom2-dependent ectodomain shedding (Figs 1A and S1) in microglia, we used primary microglia isolated from WT and iRhom2$^{-/-}$ mice. In addition, we applied CRISPR/Cas9 and generated murine microglial BV2 cells lacking iRhom2 (iRhom2$^{-/-}$) or ADAM17 (ADAM17$^{-/-}$). Compared with control BV2 cells expressing a non-targeting control (NTC) gRNA, the successful knockout of iRhom2 and ADAM17 was verified by loss of iRhom2 or ADAM17 protein in immunoblots of BV2 cell lysates (Fig 1B). Loss of ADAM17 also destabilized and reduced abundance of iRhom2, as shown before (Weskamp et al, 2020).

In agreement with previous studies (Adrain et al, 2012; McIlwain et al, 2012), the loss of iRhom2 did not affect the abundance of the proteolytically inactive proADAM17 (Fig 1B), which is found early in the secretory pathway. In contrast, the mature, proteolytically active form of ADAM17, which has a lower molecular weight because of prodomain cleavage in the TGN, was strongly reduced in iRhom2$^{-/-}$ BV2 cells (Figs 1B and S1 for a scheme). In line with the reduced abundance of mature ADAM17, the iRhom2$^{-/-}$ BV2 cells showed a strong reduction of ADAM17 activity, as measured by the release of the ADAM17 substrate TNF upon stimulation with lipopolysaccharide (LPS) (Fig 1C).

Similar to the BV2 cells, primary microglia from iRhom2$^{-/-}$ mice showed the absence of the iRhom2 protein in the cell lysates, a strong reduction of mature ADAM17, and an abolished cleavage and release of TNF upon LPS stimulation (Fig 1D and E).

Therefore, we conclude that the iRhom2$^{-/-}$ BV2 cells and primary microglia from iRhom2$^{-/-}$ mice exhibit a loss of ADAM17 function.

## iRhom2 enables ADAM17-mediated proteolysis of TREM2 in BV2 cells

To determine whether loss of iRhom2 and ADAM17 leads to a reduction of cleavage of additional membrane proteins beyond TNF, we used the "high-performance secretome protein enrichment with click sugars" (hiSPECS) method for mass spectrometry–based secretome analysis (Tüshaus et al, 2020). hiSPECS uses a metabolic labeling step with click sugars, which allows to culture cells in the presence of serum or serum-like supplements. The click chemistry reaction allows removal of abundant serum proteins, such as albumin, from the conditioned medium, while enriching the low-abundant cell secretome, which comprise soluble, secreted proteins and shed membrane protein ectodomains.

The hiSPECS analysis of the conditioned medium of iRhom2$^{-/-}$ BV2 cells revealed a greater than 50% reduction of sCSF1R (Fig 2A), a known iRhom2-dependent ADAM17 substrate (Qing et al, 2016), which validates the experimental approach for identifying ADAM17

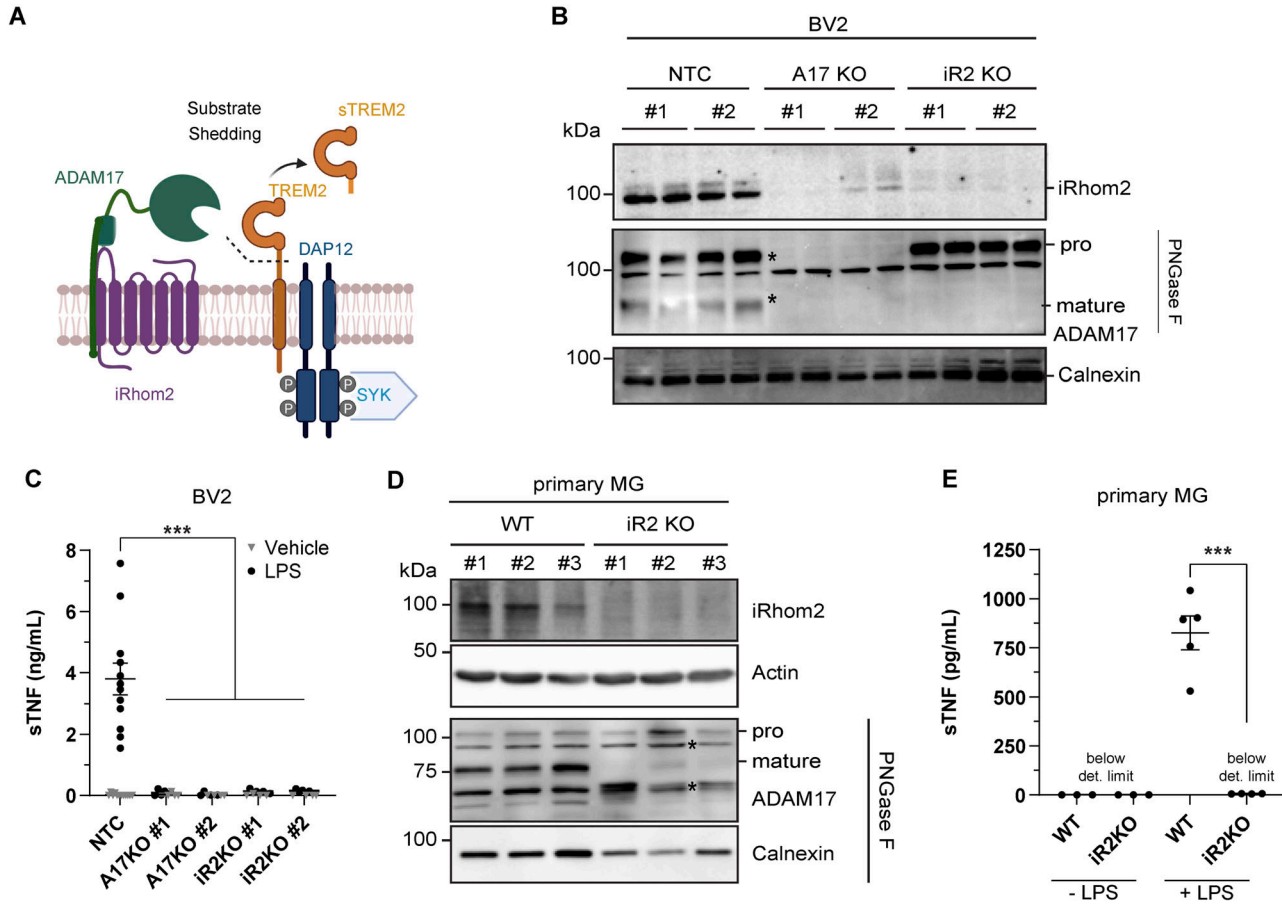

**Figure 1. Loss of iRhom2 blocks ADAM17 activity in microglia.**
**(A)** Scheme of the role of iRhom2/ADAM17 in ectodomain shedding. ADAM17 requires iRhom2 for substrate proteolysis of TREM2 at the surface of myeloid cells, resulting in release of soluble TREM2 (sTREM2). Signaling of TREM2 occurs through binding to DAP12 and subsequent phosphorylation of spleen tyrosine kinase (SYK). **(B)** Immunoblot analysis of BV2 cell lysates. Two different knockout (KO) lines were generated by single-clone expansion using CRISPR guide RNAs for both ADAM17 (A17) and iRhom2 (iR2). A non-targeting control guide RNA served as a control. Two biological replicates per single-cell clone are shown. Pro- and mature forms of ADAM17 are indicated with asterisks. To enhance separation of pro- and mature forms of ADAM17 (indicated), lysate proteins were deglycosylated with PNGase F. Calnexin served as a loading control. **(C)** Loss of ADAM17 protease activity in BV2 cells with a knockout of ADAM17 or iRhom2 (A17KO, iR2KO). Indicated BV2 KO cells were treated with LPS (100 ng/ml) for 2 h. TNF was measured in conditioned medium by ELISA (N ≥ 4). Statistical analysis was performed using a two-way ANOVA with Tukey's correction for multiple comparisons. **(D)** Immunoblot analysis of primary microglia (MG) isolated from WT or iRhom2$^{-/-}$ (iR2 KO) pups. Three biological replicates of either genotype are shown. To enhance separation of pro- and mature forms of ADAM17 (indicated), lysate proteins were deglycosylated with PNGase F. Calnexin served as a loading control. * Asterisks indicate non-specific bands. **(E)** Functional depletion of ADAM17 protease activity in iRhom2$^{-/-}$ (iR2KO) primary microglia. Primary microglia were treated or not with LPS (1 μg/ml) for 1 h. TNF was measured in the conditioned medium by ELISA (N ≥ 3). Statistical analysis was performed using a two-way ANOVA with Tukey's correction for multiple comparisons. **(C, E)** Data information: data (C, E) are represented as means ± SD from at least three independent experiments. *$P < 0.05$, **$P < 0.01$, ***$P < 0.001$. Source data are available for this figure.

substrates from iRhom2$^{-/-}$ BV2 cells. A reduction of >50% was also seen for sTREM2, for the shed ectodomains of several MHC class I proteins, including H2-K1 and H2-L, and for several additional membrane proteins (Fig 2A), and establishes them as ADAM17 substrate candidates in BV2 cells. Similar reductions of sTREM2, sCSF1R, sH2-K1, and sH2-L were also seen in ADAM17$^{-/-}$ BV2 cells (Fig 2B), revealing that their proteolytic shedding is not only dependent on iRhom2, but also on ADAM17, which is consistent with iRhom2 and ADAM17 forming the active protease complex together. A few soluble proteins were also reduced in both knockouts, including C4B and CDSN (Fig 2A and B, Table S1). Importantly, for the cleaved membrane proteins, the mass spectrometric analysis identified only tryptic peptides derived from their ectodomains, as shown for

TREM2, CSF1R, H2-K1, H2-L, MARCO, and TNFRSF1B (TNFR2) (Fig 2C), in line with the measured proteins constituting the shed ectodomains and not their full-length, transmembrane counterparts. Moreover, ADAM17 itself was reduced in the secretome of iRhom2$^{-/-}$ BV2 cells (Fig 2A). Peptides for ADAM17 were almost exclusively derived from the prodomain, which can be explained by the strongly reduced maturation and therefore prodomain release when iRhom2 is absent.

The reduction of sTREM2 in iRhom2−/− and ADAM17−/− BV2 cells, as quantified by mass spectrometry, was further validated by an ELISA in the same conditioned media, where reductions of ~90% and ~75% of sTREM2 were measured, respectively, upon loss of ADAM17 and iRhom2 (Fig 2D). The remaining sTREM2 in the knockout

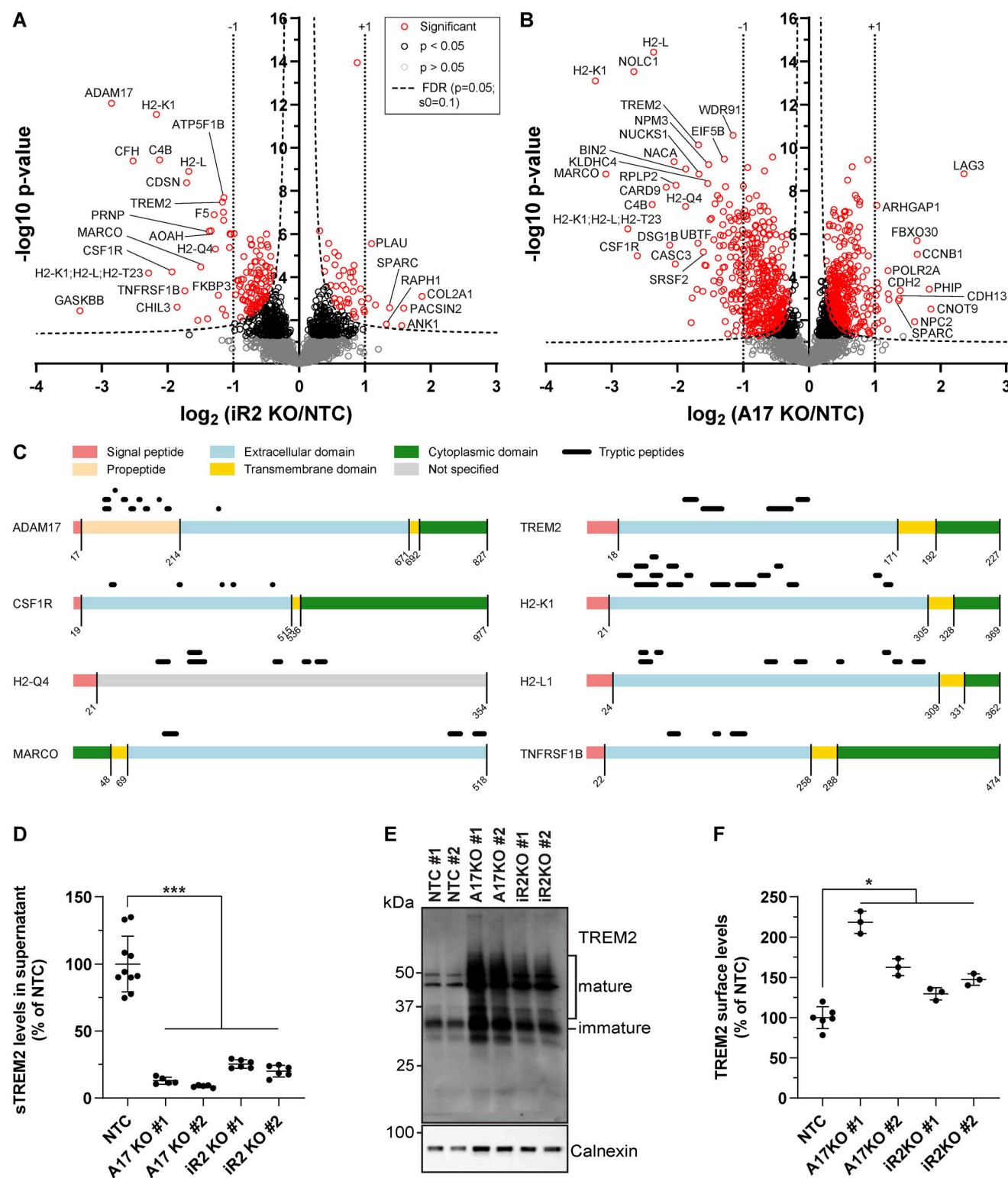

**Figure 2. iRhom2 enables ADAM17 proteolysis of TREM2 in BV2 cells.**
**(A, B)** Proteomics hiSPECS analysis comparing BV2 iRhom2$^{-/-}$ (A) or ADAM17$^{-/-}$ (B) microglia with BV2 non-targeting control (NTC) microglia. Volcano plots depict the $\log_2$ abundance change in the secretome and the negative $\log_{10}$ $P$-value for each protein (two-sample $t$ test, N = 12). The permutation-based false discovery rate estimation is represented by the black hyperbolic dashed line (FDR, $P$ = 0.05, s0 = 0.1). Vertical dotted lines indicate changes of larger or smaller than twofold in the log scale. Proteins with a $P$-value below 0.05 are highlighted with red (FDR significant) or black circles. For ease of visualization, the secretome data of the two iRhom2 (iR2$^{-/-}$: KO #1 and #2) and ADAM17 (A17$^{-/-}$: KO #1 and #2) clones were pooled and compared with the pooled secretome of the two NTC clones (NTC #1 and NTC #2). **(C)** Peptide distribution according to the specific protein domains of the indicated iRhom2/ADAM17 substrates was visualized using the web tool Quantitative Analysis of Regulated

cell supernatants may either be proteolytically generated by other proteases, such as ADAM10, or represent soluble, secreted forms of TREM2 that result from alternative splicing of the TREM2 mRNA (Del-Aguila et al, 2019; Moutinho et al, 2023). To distinguish between both possibilities, we treated the iRhom2−/− and ADAM17−/− BV2 cells with the ADAM10-preferring inhibitor GI254023X or the broader spectrum metalloprotease inhibitor BB94, which blocks ADAM10 and ADAM17, as well as a few additional metalloproteases. Treatment of the knockout cells with either GI254023X or BB94 blocked sTREM2 nearly completely (Fig S2). We conclude that sTREM2 in the BV2 cells is predominantly generated through proteolytic cleavage, mostly by ADAM17 and to a lower extent by ADAM10, rather than by alternative mechanisms, such as the generation of alternatively spliced, secreted sTREM2 variants.

Consistent with the reduced sTREM2, more full-length, transmembrane TREM2 was detected by immunoblot in the lysates of ADAM17−/− and iRhom2−/− BV2 cells (Fig 2E and quantification in Fig S3A) and this increase was also seen at the cell surface, as measured by flow cytometry (Fig 2F).

We conclude that TREM2 is an iRhom2/ADAM17 substrate in BV2 cells and that loss of iRhom2 or ADAM17 in BV2 cells results in increased amounts of full-length TREM2 in cell lysates and at the cell surface.

### iRhom2 controls TREM2 proteolysis in primary microglia

hiSPECS analysis of primary WT and iRhom2−/− microglia revealed similar iRhom2-dependent secretome changes as observed for the BV2 cells. Given the limited amount of microglia available from single mice, the total number of quantified proteins was ~770 fewer compared with BV2 cells. Likewise, fewer protein ectodomains were identified to be significantly reduced in the conditioned medium of iRhom2−/− compared with WT microglia (Fig 3A). Importantly, sTREM2 and sCSF1R showed a >50% reduction, similar to the BV2 cells. H2-L and H2-K1 also revealed diminished shedding and secretion, but the effect size was less pronounced compared with BV2 cells. The identified tryptic peptides for all three proteins mapped to their ectodomains but not their transmembrane or cytoplasmic domains (Fig 3B), consistent with their proteolytic release. Similar to iRhom2−/− BV2 cells, peptides derived from the ADAM17 prodomain, which is cleaved upon ADAM17 maturation in the TGN, were also detected to be secreted from the microglia and showed a reduction of more than 75% (Figs 2A and 3A and B). The finding of strongly reduced prodomain abundance in the media of iRhom2−/− microglia is consistent with the lack of maturation and propeptide cleavage of ADAM17 in iRhom2-deficient microglia (Fig 1D). The

reduction of sTREM2 to almost 25% upon loss of iRhom2 was validated in the same conditioned media using an ELISA (Fig 3C) and was paralleled by a selective increase in full-length, transmembrane TREM2 in the microglial lysates (Fig 3D, quantification in Fig S3B).

These results demonstrate that full-length TREM2 is enriched and sTREM2 production reduced in the absence of iRhom2 expression in primary microglia.

### iRhom2 deficiency enhances TREM2 signaling in peripheral myeloid cells

To overcome the challenge of low cell numbers obtained from primary microglial isolation, we used bone marrow–derived macrophages (BMDMs) from iRhom2−/− and WT mice to investigate whether the increased levels of full-length TREM2 in iRhom2−/− myeloid cells are associated with enhanced TREM2 signaling. Phosphorylation of the kinase SYK (pSyk) is a commonly observed downstream event in the signaling of TREM2 and additional cell surface receptors (Schlepckow et al, 2020; Wang et al, 2022; van Lengerich et al, 2023).

As expected, immunoblot analysis of iRhom2−/− BMDMs showed a lack of iRhom2 expression and impaired ADAM17 maturation (Fig 4A), consistent with the findings in BV2 cells and primary microglia (Fig 1B and D). Notably, pSyk levels were doubled in iRhom2-deficient BMDMs compared with WT controls (Fig 4A and B), consistent with the notion that the loss of iRhom2 may enhance TREM2 signaling. In addition, we observed a 50% reduction in soluble TREM2 production (Fig 4C), mirroring the results seen in BV2 cells and primary microglia (Fig 1C and E).

These findings lead us to conclude that iRhom2 controls downstream SYK signaling.

### iRhom2 deficiency enhances microglial phagocytosis and lipid droplet formation

The increased TREM2 abundance and signaling observed upon iRhom2 deficiency may be associated with TREM2-dependent changes in microglial status and activity, such as the disease-activated microglial (DAM) status, with increased phagocytic activity and altered lipid metabolism (Schlepckow et al, 2020; van Lengerich et al, 2023). We tested whether these changes in microglial activity are indeed seen in iRhom2-deficient primary microglia.

Gene expression of acutely isolated microglia was analyzed using a NanoString panel that included 235 selected genes related

---

Intramembrane Proteolysis (Ivankov et al, 2013). **(D)** Validation of iRhom2/ADAM17-mediated TREM2 ectodomain shedding in BV2 cells. Supernatant prepared for the hiSPECS analysis in (A, B) was subjected to ELISA. sTREM2 levels were quantified in conditioned media of the indicated BV2 KO cell lines (N ≥ 5). NTC (N = 10) contains the combined data of NTC #1 (N = 5) and NTC #2 (N = 5) clones. NTC was used as the baseline, and its average was normalized to 100. Statistical analysis was performed using a one-way ANOVA with Tukey's correction for multiple comparisons. **(E)** Immunoblot analysis of TREM2 in BV2 KO cell lines. Highly glycosylated, mature species of TREM2 were enriched in the ADAM17 and iRhom2 KO lines. Calnexin served as a loading control. **(F)** Surface abundance of TREM2 in BV2 KO cells. Cells were labeled with an antibody against TREM2 or isotype control to assess the geometric mean intensity of indicated BV2 KO cell lines using flow cytometry (N ≥ 3). NTC (N = 6) contains the combined data of NTC #1 (N = 3) and NTC #2 (N = 3) clones. BV2 NTC was used as the baseline, and its average was normalized to 100. Statistical analysis was performed using a one-way ANOVA with Tukey's correction for multiple comparisons. Data information: data (D, F) are represented as means ± SD from at least three independent experiments. $*P < 0.05$, $**P < 0.01$, $***P < 0.001$.
Source data are available for this figure.

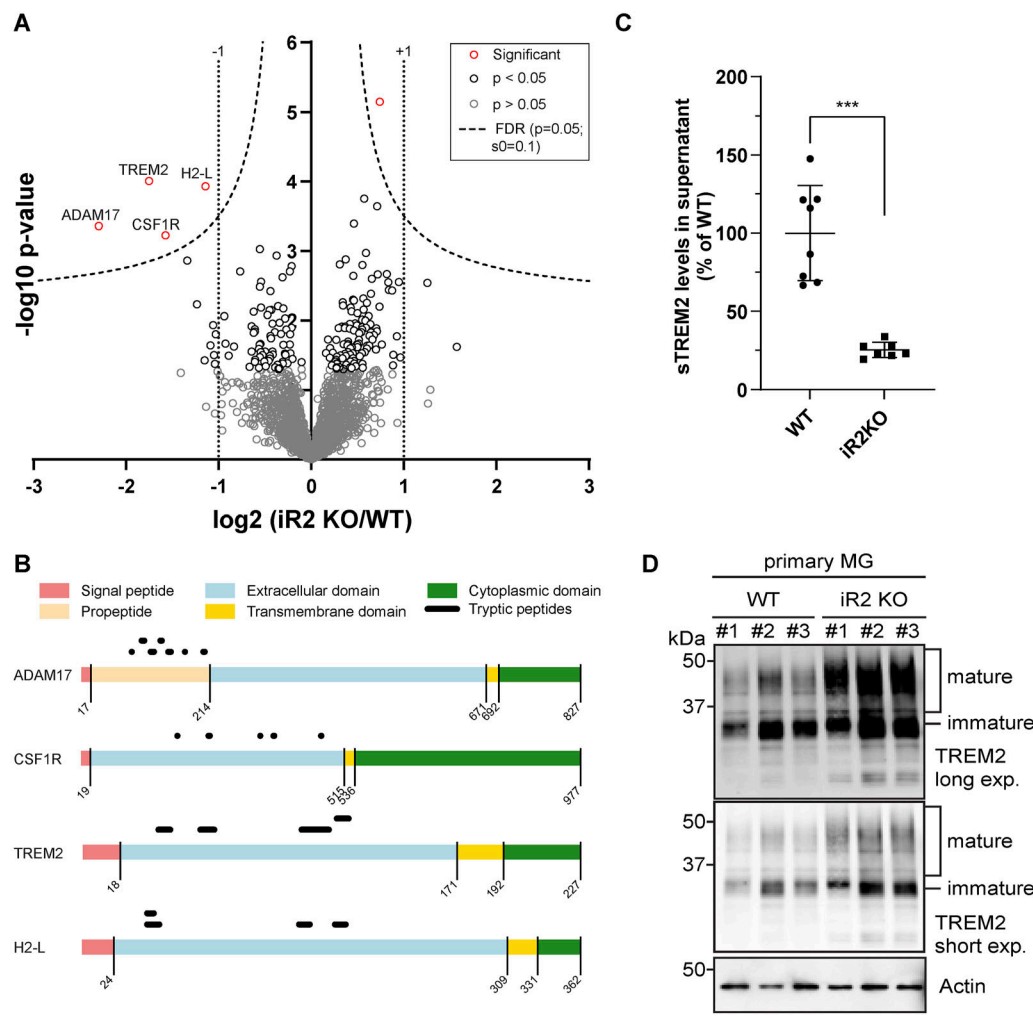

**Figure 3. iRhom2 controls TREM2 proteolysis in primary microglia.**
**(A)** Proteomics hiSPECS analysis comparing the secretome of primary microglia isolated from WT or iRhom2$^{-/-}$ (iR2 KO) pups. The volcano plot displays protein abundance changes between WT and iR2 KO supernatants. The log$_2$ fold change is plotted against the $P$-value ($-$log$_{10}$). The permutation-based false discovery rate estimation is represented by the black hyperbolic dashed line (FDR, $P$ = 0.05, s0 = 0.1). Proteins with a $P$-value below 0.05 are highlighted with red (FDR significant) or black circles. Vertical dotted lines indicate changes of larger or smaller than twofold in the log scale. **(B)** Peptide distribution according to the specific protein domains of the indicated iRhom2/ADAM17 substrates was visualized using Quantitative Analysis of Regulated Intramembrane Proteolysis (see Fig 2C). **(C)** Validation of iRhom2/ADAM17-mediated TREM2 ectodomain shedding in primary iRhom2$^{-/-}$ microglia. Supernatant prepared for the hiSPECS analysis in (A) was subjected to ELISA. sTREM2 levels were quantified in supernatants of primary microglia (N ≥ 7). A two-sided independent $t$ test was performed. **(D)** Immunoblot analysis of TREM2 levels in primary microglial lysates derived from WT or iRhom2$^{-/-}$ pups. Three biological replicates of either genotype are shown. Highly glycosylated, mature species of TREM2 were enriched in the lysates of iRhom2-deficient microglia. Actin served as a loading control. Data information: data (C) are represented as means ± SD from at least three independent experiments. *$P$ < 0.05, **$P$ < 0.01, ***$P$ < 0.001.
Source data are available for this figure.

to microglial activation, revealing an increase larger than 35% in the expression of DAM-linked genes *Clec7a, Itgax, Cst7*, and *Lpl* (Fig 5A) in iRhom2-deficient microglia. The increased expression of these genes is a hallmark of DAM, whereas genes indicative of a homeostatic signature remained unaltered (*Csf1r, Tgfbr1, P2ry12*). Yet, other genes associated with the DAM state, such as TREM2, did not show altered expression in iRhom2-deficient microglia (Fig 5A). These findings suggest that iRhom2$^{-/-}$ microglia are not in a DAM state but in a state that shares some gene expression change with the DAM state.

To investigate the phagocytic activity of iRhom2$^{-/-}$ microglia, we used an ex vivo model (Bard et al, 2000; Colombo et al, 2021), in which acutely isolated microglia are added onto Aβ plaque–bearing freshly frozen brain sections from 9-mo-old APPPS1 mice, an amyloidosis model of AD pathology (Radde et al, 2006). As a readout for microglial phagocytosis, we measured the clearance of fibrillar Aβ (Thiazin Red-positive), as determined by the area covered by the Thiazin Red staining before and after the addition of microglia (Bard et al, 2000; Colombo et al, 2021). This cleared area is normalized for the number of exogenously added microglia on the

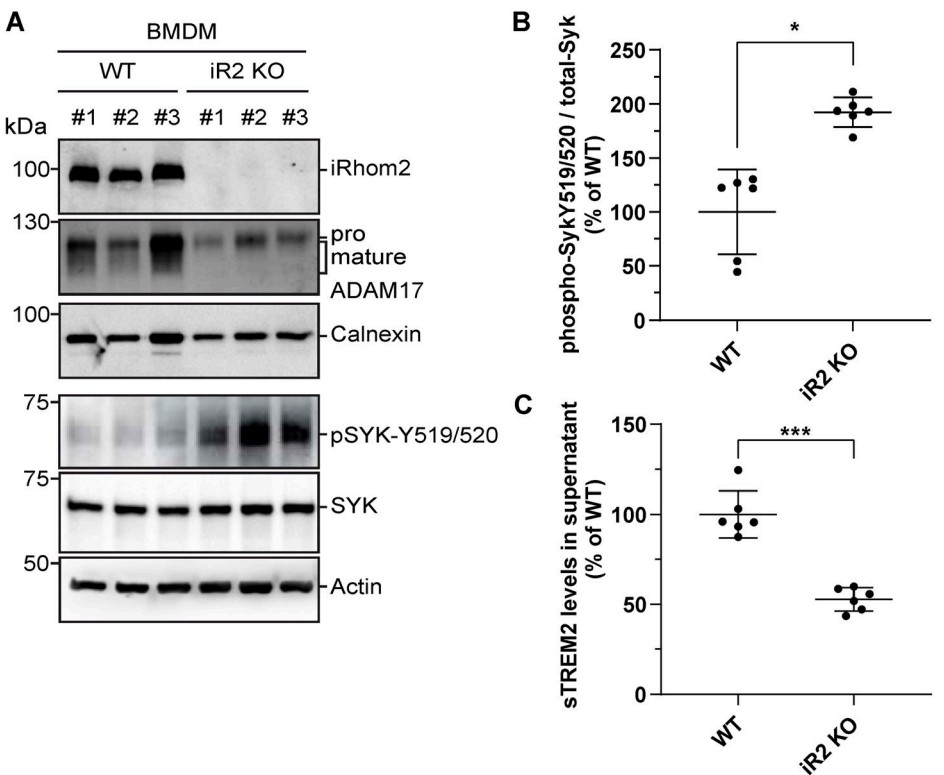

**Figure 4. iRhom2 deficiency enhances TREM2 signaling in myeloid cells.**
**(A)** Immunoblot analysis of TREM2 signaling in cell lysates isolated from WT or iRhom2$^{-/-}$ (iR2 KO) BMDMs. Three biological replicates of either genotype are shown. The membranes were probed with antibodies against iRhom2, ADAM17, or TREM2, or antibodies against spleen tyrosine kinase (SYK) or phospho-SYK-Y519/520. Calnexin and actin served as a loading control.
**(B)** Quantification of phospho-SYK-Y519/520 levels in immunoblots as shown in (A). iRhom2$^{-/-}$ BMDMs showed stronger SYK phosphorylation than WT BMDMs (N = 6). The phospho-SYK/total-SYK ratio was normalized to WT. A two-sided independent $t$ test was performed. **(C)** iRhom2/ADAM17-mediated TREM2 ectodomain shedding in iRhom2−/− BMDMs. sTREM2 levels were quantified in supernatants from cultures in (A) by ELISA (N = 6). Two-sided independent $t$ test. Data information: data (B, C) are represented as means ± SD from three independent experiments. *$P < 0.05$, **$P < 0.01$, ***$P < 0.001$.
Source data are available for this figure.

brain section, as measured by CD68 staining. iRhom2$^{-/-}$ microglia showed a 50% increase in plaque clearance efficiency compared with WT control microglia (Fig 5B), demonstrating that loss of iRhom2 enhances the phagocytic clearance activity of microglia toward Aβ.

Finally, we used the dye BODIPY to stain neutral lipids and count the number of lipid droplet–bearing CD68-positive iRhom2$^{-/-}$ versus WT microglia to detect possible changes in lipid uptake or lipid metabolism. Compared with WT microglia, the number of lipid droplet–bearing microglia was doubled in iRhom2-deficient cells (Fig 5C).

We conclude that loss of iRhom2 mildly alters the activity state of microglia, enhances their phagocytic activity toward Aβ, and increases the number of lipid droplet–bearing microglia.

## Discussion

This study identifies a physiological function of iRhom2 as a new modifier of TREM2 shedding and microglial signaling and activity and establishes the metalloprotease ADAM17, which forms a functional complex with iRhom2, as a physiological TREM2 shedding protease in microglia and BMDMs.

In our study, loss of iRhom2 in microglia and BMDMs reduced TREM2 shedding. This was accompanied by reduced sTREM2 in the conditioned medium, by increased full-length TREM2 in the cell lysates, and by enhanced TREM2 at the microglial cell surface. Consistent with the increased TREM2 surface abundance, we observed enhanced levels of pSyk, a signaling molecule activated by

TREM2 and additional cell surface receptors, elevated phagocytic activity of microglia toward Aβ, and an increase in lipid droplet–containing microglia. These cellular phenotypes are indicative of enhanced TREM2 signaling and are similar to what is seen when microglia in vitro or in vivo are stimulated with TREM2 agonistic antibodies (Cignarella et al, 2020; Schlepckow et al, 2020; Wang et al, 2020; van Lengerich et al, 2023). These antibodies bind TREM2 close to the proteolytic cleavage site, cross-link TREM2 and block TREM2 shedding, increase surface TREM2, reduce sTREM2, activate pSyk signaling, and mediate enhanced Aβ phagocytosis and the expression of the DAM marker Itgax.

Changes in TREM2 shedding are genetically linked to AD, but it remains unclear whether increased TREM2 cleavage is beneficial or detrimental for AD. On the one hand, the TREM2 H157Y mutation, which is located directly at the TREM2 cleavage site, enhances TREM2 shedding, reduces cell surface TREM2, lowers TREM2-dependent phagocytosis, and is associated with late-onset AD (Jiang et al, 2016; Feuerbach et al, 2017; Schlepckow et al, 2017; Thornton et al, 2017; Fu et al, 2023; Qiao et al, 2023). Consequently, several agonistic TREM2-targeted antibodies are in preclinical and clinical development for AD, which bind TREM2 close to its cleavage site and reduce TREM2 shedding (Wang et al, 2020; Schlepckow et al, 2023; van Lengerich et al, 2023). On the other hand, genetic studies demonstrated that increased sTREM2 in CSF is associated with reduced AD risk. Specifically, genetic variants of the MS4A gene cluster and, more recently, of TGFBR2 and NECTIN2 are associated with increased CSF sTREM2 and reduced AD risk (Deming et al, 2019; Wang et al, 2024). Moreover, sTREM2 has protective functions (Wu et al, 2015; Zhong et al, 2017; Zhong et al, 2019), for example, when

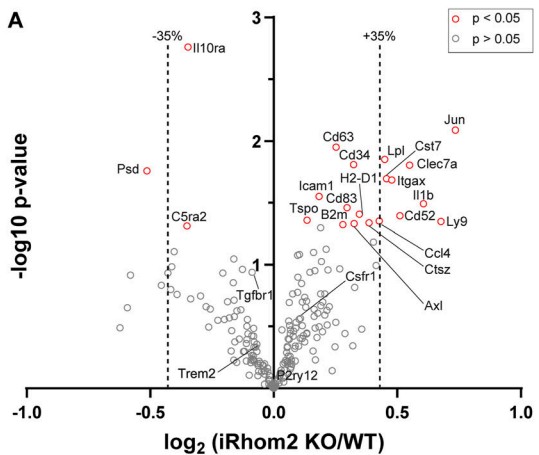

**A**

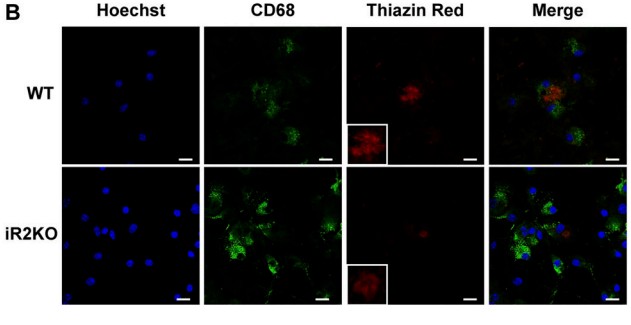

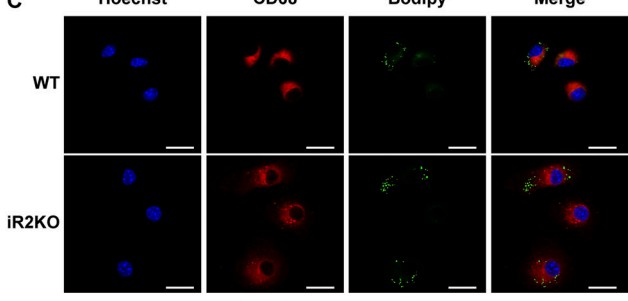

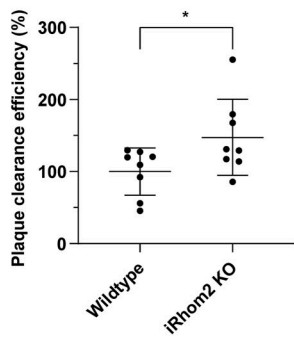

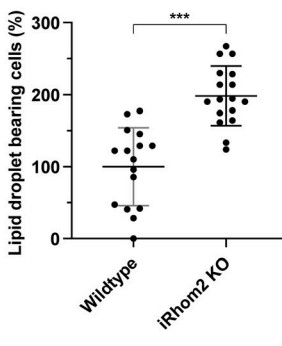

**Figure 5. iRhom2 deficiency enhances microglial phagocytosis and lipid droplet formation.**
**(A)** Quantitative analysis of microglial gene expression. The volcano plot illustrates the transcriptomic changes in acutely isolated microglia from adult iRhom2$^{-/-}$ (iR2 KO) mice compared with WT controls (N ≥ 6). The x-axis shows the log$_2$ fold change, whereas the y-axis represents the P-value (-log$_{10}$). Genes with a change greater than 35% (dashed vertical line) and a P-value less than 0.05 (represented as red circles) were considered significantly altered. **(B)** Phagocytosis of amyloid-beta (Aβ) plaques is increased in iRhom2$^{-/-}$ microglia. Representative images of acutely isolated microglia from WT or iRhom2$^{-/-}$ mice plated onto APPPS1 cryosections containing Aβ plaques. Microglial lysosomes were stained with an antibody against CD68 (green), fibrillar Aβ plaques with Thiazine Red (ThR, red), and nuclei with Hoechst (blue). Scale bar: 20 µm. Inset: magnification of the plaque area. The phagocytic activity is measured as plaque clearance efficiency by comparing the ThR-positive area in a brain section incubated with microglia to that in the consecutive brain section without added cells (N = 8). Data are normalized to WT. Two-sided independent t test. **(C)** Elevated numbers of lipid droplet–bearing cells in iRhom2$^{-/-}$ primary microglia. Representative images of WT or iRhom2$^{-/-}$ microglia plated on coverslips. At four days in vitro (DIV4), microglial lysosomes were stained with CD68 (red), lipid droplets with BODIPY (green), and nuclei (blue) with Hoechst. Scale bar: 20 µm. Lipid droplet–bearing cells were counted (N ≥ 16). Two-sided independent t test. Data information: data (B, C) are represented as means ± SD from at least three independent experiments. *P < 0.05, **P < 0.01, ***P < 0.001. Source data are available for this figure.

injected into the brains of AD mice. At present, it remains unclear why increased TREM2 cleavage and sTREM2 production are beneficial in some studies but detrimental in other studies. Potentially, the beneficial versus detrimental effects depend on whether the corresponding study investigates effects specifically on amyloid or tau pathology in vitro or in animal models (Sayed et al, 2018; Haass, 2021; Jain et al, 2023) or on AD risk in general, as done in genetic studies (Deming et al, 2019; Wang et al, 2024).

The metalloproteases ADAM10, ADAM17, and meprin β were shown to cleave TREM2 in cell lines (Feuerbach et al, 2017; Schlepckow et al, 2017; Thornton et al, 2017; Berner et al, 2020), but the physiological TREM2-cleaving protease was previously unknown in microglia. Our study establishes the ADAM17/iRhom2 protease complex as a TREM2 protease in microglia and BMDMs. Yet, the knockout of iRhom2 or ADAM17 did not fully abolish sTREM2, suggesting that one or several

additional proteases may also contribute to TREM2 shedding. In fact, our pharmacological experiments demonstrate that ADAM10 also contributes to TREM2 proteolysis in BV2 cells, albeit to a lower extent than ADAM17. This result is in line with a recent study demonstrating that TREM2 shedding in human iPSC-derived microglia partly depends on TSPAN14 (Yang et al, 2023), a recently identified AD risk gene, which is suggested to act as a non-proteolytic subunit of an ADAM10 protease complex (Kleinberger et al, 2014; Harrison et al, 2021; Schwartzentruber et al, 2021; Bellenguez et al, 2022; Yang et al, 2023). It is not unusual that a membrane protein undergoing shedding can be cleaved by both ADAM10 and ADAM17 and this is also known for the AD-linked amyloid precursor protein (APP) (Saftig & Lichtenthaler, 2015; Jocher et al, 2022).

iRhom2 is a member of the rhomboid family of intramembrane serine proteases, but is catalytically inactive, as it lacks the active

site residues required for serine protease activity. Yet, iRhom2 is involved in proteolysis, as it forms a complex with ADAM17 and controls the life cycle and activity of ADAM17 (Adrain et al, 2012; McIlwain et al, 2012). ADAM17 cleaves TNF and IL-6 receptor and is a central drug target for inflammatory conditions, such as sepsis and rheumatoid arthritis. ADAM17 is also required for cleavage and signaling of several EGFR ligands and for tissue homeostasis (Blobel, 2005; Zunke & Rose-John, 2017). Given these multiple functions and the close homology between ADAM17 and ADAM10, selective and safe inhibition of ADAM17 with small-molecule drugs was not successful (Moss et al, 2008). Inhibition of iRhom2 appears more promising. The global loss of iRhom2 induces a myeloid cell–specific loss of ADAM17 activity, given that in tissues and cell types other than myeloid cells, the iRhom2-homolog iRhom1 can compensate for the loss of iRhom2 and support the functions of ADAM17, for example, in skin and intestinal barrier protection.

Increased iRhom2 expression is associated with enhanced AD risk (De Jager et al, 2014; Lunnon et al, 2014; Lardenoije et al, 2019; Li et al, 2020; Palma-Gudiel et al, 2023) and may lead to augmented iRhom2-dependent TREM2 shedding. Thus, it is conceivable that an iRhom2-targeted, selective ADAM17 inhibition may also be tested for AD and may alleviate amyloid pathology in a manner similar to what is achieved with anti-TREM2 cleavage site antibodies that block its proteolytic release.

In summary, our study identifies ADAM17 as a physiological protease for TREM2 in microglia, establishes iRhom2 as a modifier of TREM2 shedding, and, thus, elucidates a mechanism by which iRhom2 may contribute to the biology underlying AD.

# Materials and Methods

### Mouse work

All mouse work was conducted in accordance with the European Communities Council Directive, and animal experiments were approved by the committee responsible for animal ethics of the government of Upper Bavaria (license: #02-19-067). The mice were bred and housed in a pathogen-free animal facility at the German Center for Neurodegenerative Diseases (DZNE) in Munich. The $Rhbdf2^{-/-}$ mice were obtained from Dr. Tak W. Mak at UHN (international designation: B6.129-iRhom2tm1Mak). APPPS1 mice (C57BL/6J-Tg(Thy1-APPSw,Thy1-PSEN1*L166P)21Jckr/J) were obtained from Mathias Jucker (Radde et al, 2006). The targeting strategy and characterization of this line have been previously described (McIlwain et al, 2012). For isolation of $Rhbdf2^{+/+}$ and $Rhbdf2^{-/-}$ microglia, heterozygous $Rhbdf2^{+/-}$ mice were mated to compare $Rhbdf2^{-/-}$ with $Rhbdf2^{+/+}$ littermate controls.

### Reagents and antibodies

The rabbit polyclonal anti-ADAM17 cytotail antibody used for Western blotting was described previously (Schlöndorff et al, 2000). The rat monoclonal antibody against the cytoplasmic domain of iRhom2, used at 1:10, has been previously described (Weskamp et al, 2020). The rabbit polyclonal anti-alpha-tubulin antibody was purchased from Abcam (1:2,000, ab4074). The mouse monoclonal antibody anti-beta-actin was obtained from Sigma-Aldrich (1:2,000, A5316). The rabbit polyclonal anti-calnexin antibody was purchased from Enzo (1:2,000, ADI-SPA-860-F). The rat anti-TREM2 (5F4) antibody, used at 1 μg/ml, for Western blotting and as the detection antibody for ELISA was described previously (Schlepckow et al, 2020). The biotinylated sheep anti-mouse TREM2 from Bio-Techne (0.125 μg/ml, BAF1729) was used as the capture antibody. The secondary anti-rat Sulfo-tag was purchased from MesoScale (1:1,000, R32AH-5). For detecting TREM2 on the cell surface using flow cytometry, the monoclonal anti-mouse TREM2 APC-conjugated antibody from Bio-Techne was used (0.4 μg/106 cells, FAB17291A). The rat IgG2b APC-conjugated isotype control antibody was from Bio-Techne (0.4 μg/106 cells, IC013A). For detection of Syk, a rabbit anti-Syk (2712; Cell Signaling) was used at 1:1,000, and phospho-Syk was detected using rabbit anti-phospho-Syk (C87C1) at 1:1,000 (2710; Cell Signaling). For immunofluorescence stainings, the monoclonal anti-mouse CD68 was from Bio-Rad (1:500, #MCA1957). Fibrillar Aβ (plaque core) was visualized using Thiazine Red from Sigma-Aldrich (2 μM, CDS012020). BODIPY 493/503 dye (D3922) and Hoechst 33342 (H3570) were obtained from Thermo Fisher Scientific. The cells were stimulated with LPS at 100 ng/ml (L4391; Sigma-Aldrich), GM-CSF at 20 ng/ml (415-ML-020; R&D), or M-CSF at 50 ng/ml (416-ML-050; R&D). Tetra-acetylated N-azidomannosamine (ManNAz) was obtained from Thermo Fisher Scientific (50 μM, C333366). The ADAM10-preferring inhibitor GI254023X from Sigma-Aldrich (SML0789) was used at 10 μM, and the broad-spectrum matrix metalloprotease inhibitor BB94 (Batimastat) from SelleckChem was used at 10 μM concentration.

### Cell culture

Cell lines and primary cells were cultured under standard conditions at 37°C and 5% $CO_2$. Cells were maintained with DMEM, DMEM/F12, or RPMI medium supplemented with 1% penicillin–streptomycin and 10% FBS.

### Generation of BV2 CRISPR clones

For CRISPR/Cas9-mediated knockout of $Rhbdf2$/iRhom2 and ADAM17, the pL-CRISPR.EFS.GFP plasmid, which was a gift from Benjamin Ebert (plasmid # 57818; Addgene; http://n2t.net/addgene:57818; RRID:Addgene_57818), was used, as previously described (Heckl et al, 2014). The guide RNAs (gRNAs) were selected using online tools to minimize the risk of off-target effects (https://zlab.squarespace.com/guide-design-resources, https://portals.broadinstitute.org/gppx/crispick/public). Guide sequences for ADAM17 (guide #1: 5′-CCTCTCTTCTCTCCACACTTC-3′ [Exon 1], guide #2: 5′-TAGCTAATATTCAGCAGCAC-3′ [Exon 2]), RHBDF2/iRhom2 (guide #1: 5′-AAAACCGTGCAAGATGCCCA-3′ [Exon 5], guide #2: 5′-CCCTCCTGCCATCGTCCCCG-3′ [Exon 4]), and a NTC control sequence (NTC guide #1, 5′-GCGAGGTATTCGGCTCCGCG-3′, NTC guide #2: 5′- GATCTTG-GATCTTAGCTGCG-3′) were cloned into the pL-CRISPR.EFS.GFP vector.

To generate CRISPR/Cas9-mediated knockout lines, the pL-CRISPR.EFS.GFP plasmid containing gRNA was transduced into BV2 using a lentivirus. The lentivirus was packaged and purified as described previously with slight modifications (Jocher et al, 2022). Briefly, HEK293T cells were plated at a density of $1.6 \times 10^7$ cells per

15-cm dish to package the CRISPR constructs into the virus. The cells were incubated overnight at 37°C and 5% CO$_2$ in DMEM (10% FBS, 1% P/S). For transfection, 2 ml of Opti-MEM was mixed with 80 $\mu$l of Lipofectamine 2000, and in a separate tube, 2 ml of Opti-MEM was mixed with 60 $\mu$g psPAX2, 40 $\mu$g pcDNA3.1-VSVG, and 80 $\mu$g of the respective CRISPR/Cas9-eGFP/gRNA construct. These mixtures were combined and left at RT for an hour. One hour before transfection, the medium was replaced with 16 ml of Opti-MEM supplemented with 10% FBS. The transfection mix was then added dropwise, and the cells were incubated overnight. The next day, the medium was replaced with 20 ml of DMEM containing 2% FBS, 1% P/S, and 10 mM sodium butyrate. After overnight incubation, the supernatants were centrifuged at 1,000$g$ for 10 min and filtered through a 0.45-$\mu$m filter. The supernatant was then centrifuged at 22,000 rpm (87,300$g$) in a SW28 rotor for 3 h at 4°C. The pellet was resuspended in 200 $\mu$l of TBS-5 (TBS + 5% BSA) buffer and incubated at 4°C for 4 h before the viral particles were aliquoted into small PCR tubes and frozen at −80°C.

BV2 cells were seeded at a density of 6 × 10$^5$ in the presence of 10 $\mu$l of the purified lentiviral particles. After 24 h, eGFP-expressing BV2 cells were sorted into 96-well plates using FACSMelody (BD). The growth of single-cell clones was constantly observed by microscopy. After reaching confluence, the cells were detached by trypsin/EDTA and transferred to a T175 flask. Single-cell clones were tested for successful KO using sequencing. Genomic DNA was isolated by extraction in a lysis buffer (100 mM Tris, pH 8.5, 5 mM EDTA, 0.2% SDS, 200 mM NaCl, 100 $\mu$g/ml proteinase K freshly added), followed by centrifugation (15 min at 16,000$g$ at 4°C), and subsequent precipitation using 40% (vol/vol) isopropanol, and two washing steps with 70% (vol/vol) ethanol. Genomic DNA was recovered by drying and resuspension in distilled H$_2$O. To determine the modifications introduced by CRISPR/Cas9, gRNA-targeted regions were amplified by PCR. The amplicon was verified and purified by agarose gel electrophoresis and then subjected to Sanger sequencing (Eurofins Genomics).

For PCR amplification and sequencing, the following primers were used, with one primer pair per guide: for ADAM17, forward guide #1: 5'-AGGTTTCCCAGAGAGGTGGT-3', reverse guide #1: 5'-AGTTTGCTCCCCGAACTCTT-3'; forward guide #2: 5'-TCAGCCTTCG-TAATCTCTCC-3', reverse guide #2: 5'-CCAAAACAAAACCCACCCC-3'; and for RHBDF2, forward guide #1: 5'-GCCCACACCGTATCTGTTCT-3', reverse guide #1: 5'-CTGCAAGAGATGTGGGTGAA-3'; forward guide #2: 5'-CAGGAACCCAGGGCTTTAGG-3', reverse guide #2: 5'-TTCTGGCCTTC CACATCCAC-3'.

The obtained sequences were analyzed with the help of Synthego's Inference of CRISPR Edits tool (https://ice.synthego.com/). Validation of successful gene knockout on the protein level was achieved by Western blotting as mentioned below.

## Isolation of primary microglia

Microglial isolation was performed using a standard operating procedure based on the neuronal dissociation kit (P) and magnetic cell separation (MACS) technology from Miltenyi Biotec (Lee et al, 2008). Briefly, after dissecting the brain, the cerebellum, olfactory bulb, and brainstem were removed. The remaining cerebrum was enzymatically dissociated, and microglia were isolated by magnetic

selection of CD11b+ cells using a MACS separation column. The isolated microglia were centrifuged for 5 min at 300$g$, then either snap-frozen in liquid nitrogen or resuspended in culture media, depending on the experimental requirements.

## Isolation and differentiation of bone marrow–derived macrophages

BMDMs were isolated from adult mice older than 8 wk as described previously (Weskamp et al, 2020). Briefly, bone marrow was extracted from the femurs and tibiae of adult mice using Hanks' Balanced Salt Solution (HBSS, 14025100; Life). The bones were flushed using a syringe containing advanced DMEM (61965059; Thermo Fisher Scientific) to extract bone marrow. The resulting cell suspensions were passed through a 100-$\mu$m cell strainer and then incubated for 2 min in ACK lysis buffer (A1049201; Thermo Fisher Scientific) to lyse red blood cells. The extracted marrow was centrifuged at 300$g$ for 10 min to collect the cells. These cells were then differentiated in DMEM with 10% FBS (AC-SM-0027; Anprotec), 1% penicillin–streptomycin, and 50 ng/ml murine macrophage colony-stimulating factor (M-CSF) on a 10-cm petri dish. After 7 d in vitro, the cells were detached using Accutase (SCR005; Millipore) treatment and gentle scraping. Mature BMDMs were then cultured at a density of 1.5 × 10$^6$ cells per six-well plate. Supernatants were collected, and cell lysis was performed after 24 h.

## SDS–PAGE and Western blotting

Isolated or cultured cells were washed twice with PBS to remove residual culture media before lysis with STET buffer (50 mM Tris, 150 mM NaCl, 2 mM EDTA, 1% Triton, pH 7.5) containing protease inhibitor cocktail (1:500) and 10 mM 1,10-phenanthroline to inhibit ADAM17 autoproteolysis (Schlöndorff et al, 2000). The protein concentration was measured using Pierce BCA Protein Assay Kit. Samples were typically boiled for 10 min at 95°C in 1x reducing Laemmli buffer (4x stock: 8% SDS, 40% glycerol, 0.025% bromo-phenol blue, 10% $\beta$-mercaptoethanol, 125 mM Tris, pH 6.8). To detect iRhom2, boiling was omitted. Samples were separated on 8–12% Tris-glycine gels in a Mini-PROTEAN tetra cell (Bio-Rad). Proteins were transferred to 0.2-$\mu$m nitrocellulose membranes using the Trans-Blot Turbo system (Bio-Rad). Membranes were blocked for 1 h in PBS-T with 5% skim milk powder, then incubated overnight at 4°C with primary antibodies prepared in 5% BSA in PBS-T. After washing three times with PBS-T, membranes were incubated with HRP-conjugated secondary antibodies for 45 min at RT. Detection was performed using ECLTM (GERPN3243; Sigma-Aldrich), and images were recorded on the ImageQuant LAS 4000 platform. Images were edited with Photoshop (version 12.1). Densitometric quantifications were done using Fiji (ImageJ 2.0.0-rc-67/1.52c) or ImageLab 3.0 software. Changes in the knockout cells are given as fold-change ratios relative to the control cells (WT or NTC). In Fig S3A, the TREM2 intensities were in addition normalized by the intensity of the loading control housekeeping protein.

For detection of phospho-Syk, the lysis buffer contained PhosSTOP™ reagent (4906845001; Sigma-Aldrich). All blocking and washing steps were performed in the presence of TBS/TBS-T.

To enhance the separation of the pro- and mature forms of ADAM17 via Western blotting, the cell lysate was treated with PNGase F (P0709S; NEB) according to the manufacturer's instruction. Briefly, samples were boiled at 95°C for 10 min. Digestion was completed by adding Glycobuffer 2, NP-40, and PNGase F and incubated at 37°C for 1 h. Laemmli buffer was added, and samples were boiled at 95°C for 5 min and run on 8% SDS–PAGE.

To optimize the detection of TREM2 in cell lysates, membranes were enriched using a previously described procedure (Kleinberger et al, 2014).

## ELISA

To detect murine soluble TNF in culture supernatants, primary microglia were seeded at $1 \times 10^6$ cells/well in a six-well plate and cultured in DMEM/F12 (10% FBS, P/S). At DIV3, the media were replaced with DMEM/F12 (1% Pen/Strep) containing 1 µg/ml LPS or PBS as a control for 1 h. BV2 cells were seeded at $3 \times 10^5$ cells/well in a six-well plate and stimulated the next day with 100 ng/ml LPS or PBS for 2 h. After LPS stimulation, the media were collected and centrifuged at 3,000$g$ for 5 min. TNF levels in the supernatants were measured using Mouse TNF-alpha Tissue Culture Kit from MSD according to the manufacturer's instructions.

To detect murine sTREM2 in culture supernatants of primary microglia, BV2 cells, and BMDM, we used a custom protocol based on the Small Spot Streptavidin Sector GOLD technology (MSD), as detailed elsewhere (Kleinberger et al, 2014). Briefly, MSD GOLD plates precoated with streptavidin were blocked overnight at 4°C with blocking buffer (PBS, 0.05% Tween, 3% BSA). After discarding the blocking buffer, plates were incubated with 25 µl of a biotinylated anti-TREM2 antibody (BAF1729; R&D Systems) at 0.125 µg/ml for 90 min at RT. Plates were then washed three times with a washing buffer (PBS, 0.05% Tween).

Culture supernatants were diluted 1:40 in PBS with 0.05% Tween, 1% BSA, and protease inhibitor cocktail. 50 µl of diluted samples was added to the wells.

A standard curve was created using a twofold serial dilution of recombinant mTREM2-FC (200 ng/ml stock; R&D) in PBS, ranging from 400 pg/ml to 12.5 pg/ml. To prevent FC-tagged TREM2 dimerization, the standard samples were denatured by the addition of denaturing buffer (200 mM Tris–HCL, pH 6.8, 4% SDS, 40% glycerol, 2% ß-ME, 50 mM EDTA) and boiling at 95°C for 5 min 50 µl of each standard and supernatant sample was dispensed into the wells in duplicates. Samples and standards were incubated for 2 h at RT.

Plates were washed and incubated with rat anti-TREM2 detection antibody (clone 5F4, stock: 1 mg/ml; Sigma-Aldrich) at 1:1,000 dilution for 1 h. After washing, a goat anti-rat Sulfo-tag secondary antibody (MSD) at 1:1,000 dilution was added and incubated for 1 h. Plates were washed again, and 150 µl of 1x read buffer (MSD) was added. The plates were read using the MSD platform, and TREM2 levels were determined using a four-parameter logistic fit curve regression model.

## Mass spectrometry

### hiSPECS analysis
Secretome analysis of cultured primary microglia or BV2 cells was conducted using the high-performance secretome protein enrichment with click sugars (hiSPECS) method as previously described (Tüshaus et al, 2020) with minor modifications. Briefly, heterozygous WT/iR2KO animals at 3 mo old were bred, and their genotypes were confirmed using PCR. The offspring were derived from three separate mating pairs, each producing at least one pup of each genotype. Microglia were harvested at postnatal day 9 (p9) as described above, yielding a minimum of one million cells per brain. $1 \times 10^6$ CD11b+ microglia were placed in a six-well plate and cultured for 3 d. After this period, the culture medium was replaced with DMEM/F12 (10% FBS, 1% P/S) containing 50 µM ManNAz. The media were collected after 72 h, briefly filtered by centrifugation through a 0.22-µm Costar Spin-X column (CLS8160; Merck), and stored at −20°C. For hiSPECS analysis of BV2 cells, $1 \times 10^6$ cells were cultured in RPMI (10% FBS, 1% P/S) in six-well plates. The next day, the media were exchanged with fresh media containing 50 µM ManNAz. The supernatants were collected after 48 h and filtered and stored as mentioned above.

The supernatants were thawed and subjected to a lectin-based enrichment using concanavalin A (ConA) (C7555; Sigma-Aldrich) and wheat–germ agglutinin (WGA) (C7555; Sigma-Aldrich). After elution, the proteins were bound to magnetic dibenzocyclooctyne (DBCO) beads (CLK-1037; Jena Bioscience) via click chemistry. Stringent washing of the magnetically immobilized glycoproteins removed contaminants. The glycoproteins were then eluted, enzymatically digested with trypsin and LysC, and subjected to SP3 contaminant removal.

### SP3 contaminant removal
The single-pot, solid-phase–enhanced sample preparation (SP3) technique was used to remove contaminants and exchange the liquid matrix. The employed protocol is based on the original protocol from Hughes et al (2019) with modifications (Hughes et al, 2019). The contaminant-free tryptic peptides were resolved in 0.1% FA, briefly sonicated, and then subjected to LC-MS/MS.

### LC-MS/MS analysis
Samples were analyzed using liquid chromatography–mass spectrometry (LC-MS/MS) with a nanoElute nanoHPLC system connected to a timsTOF Pro mass spectrometer (Bruker, Germany). Eight microliters of each sample was separated on a nanoElute nanoHPLC system equipped with an in-house–packed C18 analytical column (30 cm × 75 µm ID, ReproSil-Pur 120 C18-AQ, 1.9 µm; Dr. Maisch GmbH). The chromatography employed a binary gradient of water (solvent A) and acetonitrile (solvent B), both containing 0.1% formic acid, at a flow rate of 300 nl/min. The column temperature was maintained at 50°C. The gradient was as follows: 0 min, 2% B; 3.5 min, 5% B; 48 min, 24% B; 59 min, 35% B; 64 min, 60% B.

Samples were analyzed using Data-Independent Acquisition (DIA) parallel accumulation serial fragmentation. For the microglial hiSPECS analysis, each scan cycle included one MS1 full scan followed by two rows of 48 sequential DIA windows, each with a 20-m/z width and a 1-m/z overlap, covering a scan range of 300–1,200 m/z. The ramp time was set to 166 ms, and six windows were scanned per ramp, resulting in a total cycle time of 2.8 s. For microglial lysates, each scan cycle included one MS1 full scan followed by two rows of 42 sequential DIA windows, each with a 23-m/z width and a 1-m/z overlap, covering a scan range of 350–1,275 m/z. The ramp time was

set to 120 ms, with six windows scanned per ramp, leading to a total cycle time of 1.8 s.

### LC-MS/MS data analysis

The MS raw data analysis was conducted using DIA-NN software, version 1.81, via a library-free search as described earlier (Demichev et al, 2020). For this analysis, a canonical *Mus musculus* database, which includes one protein per gene, was used. This database was obtained from UniProt and contained 21,966 entries as of 9 April 2021. Trypsin was specified as the protease with cleavage specificity at the C terminal of K and R. Variable modifications included acetylation at protein N termini and oxidation of methionines, whereas carbamidomethylation of cysteines was set as a fixed modification. The software automatically adjusted mass tolerances for peptide and fragment ions, as well as ion mobility tolerances. The analysis allowed for two missed cleavages. The false discovery rate (FDR) was set at 1% for both proteins and peptides. For quantification, only unique and razor peptides were considered. Protein LFQ intensities were accepted for two peptide identifications.

The proteomics data analysis was performed in Perseus v1.6.2.3 (Tyanova et al, 2016) and Excel 2019. To evaluate protein abundance changes, protein LFQ intensities were $\log_2$-transformed. Statistical evaluation required at least three valid LFQ values per group. A two-sided *t* test was applied to check for significant changes between the groups. To account for multiple hypotheses, a permutation-based FDR correction was applied (Tusher et al, 2001). The analyzed proteomics data are listed in Table S1.

### Flow cytometry

For quantifying TREM2 surface levels, BV2 cells were seeded ($5 \times 10^5$) in 12-well plates. The next day, cells were thoroughly detached by pipetting and incubated on ice for 15 min in flow cytometry buffer (PBS, 1% BSA, 0.25% $NaN_3$) containing 1% mouse SeroBlock FcR (BUF041A; Bio-Rad). An APC-conjugated anti-TREM2 antibody (FAB17291A; R&D) or an isotype control antibody (IC103A; R&D) was used to stain surface TREM2 by incubation for 30 min on ice. Cells were washed three times. Flow cytometry analysis was performed using BD FACSMelody, and data were analyzed using FlowJo software (v.10.4.1). A total of 10,000 events were recorded per sample, whereas doublets and dead cells were excluded from analysis by gating and propidium iodide (P4864; Sigma-Aldrich) staining, respectively.

### NanoString

Primary microglia were isolated from adult (22-wk-old) *WT* and $Rhbdf2^{-/-}$ animals using MACS as described before. Total RNA was extracted using the miRNeasy kit (QIAGEN) according to the manufacturer's instructions. All RNA samples had RNA Integrity (RIN) scores above 7.0, determined using the Eukaryote Total RNA Nano Series II assay on Agilent 2100 Bioanalyzer. cDNA was synthesized by reverse transcription, and a multiplexed target enrichment was performed on 13 samples (6x WT, 7x KO).

RNA levels of a customized codeset containing 226 genes were analyzed using NanoString technology at Proteros. Data analysis

was performed using Gene Expression nCounter RCC analysis software by ROSALIND (version 2.2.3.2). Low expressors were pruned if the transcript was below the background threshold in more than 85% of the samples, leading to the exclusion of seven genes (Arg1, C3, Nos2, Igf1, Il10, Ch25h, and Pparg). Consequently, 218 genes were included in the differential gene expression analysis. Genes were considered differentially expressed if the *P*-value was less than 0.05. Statistical analyses were conducted in the ROSALIND environment. The extent of gene differential expression between experimental conditions was visualized using GraphPad Prism 10.

### Ex vivo Aβ plaque clearance

To evaluate the phagocytic clearance of Aβ by primary microglia, we used an ex vivo assay similar to a phagocytic assay as previously described (Bard et al, 2000; Colombo et al, 2021).

In brief, freshly frozen 10-$\mu$m-thick brain sections of brain from 9-mo-old APPPS1 mice (C57BL/6J-Tg(Thy1-APPSw,Thy1-PSEN1*L166P) 21Jckr/J) (Radde et al, 2006) were positioned on a glass coverslip coated with poly-L-lysine. To enhance cell migration to the plaque area, the brain sections were treated for 1 h at RT with 5 $\mu$g/ml of an anti-human Aβ antibody (6E10, 803015; BioLegend). Isolated microglia were seeded onto coverslips containing brain sections at a density of $3 \times 10^5$ cells and cultured for 5 d in their respective media before being fixed with 4% PFA/sucrose for 15 min at RT. Immunostainings were performed using anti-CD68, Hoechst, and ThR to visualize microglia, nuclei, and fibrillar Aβ in the plaque cores, respectively. ThR was included in the secondary antibody mixture. Images were recorded using a Leica SP5 confocal microscope. The phagocytic capacity was quantified by measuring the coverage of plaques (ThR signal area) comparing sections incubated with cells to a consecutive brain section without cells. Analysis was conducted with a macro in Fiji (ImageJ 2.0.0-rc-67/ 1.52c), using a threshold algorithm (OTSU) to analyze 10 × 16-bit scans of the entire coverslip, focusing on particles ranging in size from five pixels to infinity. This procedure was replicated in three independent experiments. The cleared Aβ area (ThR staining) was then normalized for the number of exogenously added microglia on the brain section, as measured by CD68 staining. To ensure that only the exogenously added microglia, but not the microglia within the brain section, were detected by CD68 staining, the fixed slices were permeabilized for only 3 min, which permeabilizes only microglia at the surface of the slice.

### Lipid droplet staining in primary microglia

Primary microglia from p9 old pups (P9) were isolated as described before and seeded at a density of $5 \times 10^4$ into the $\mu$-Slide eight-well Ibidi chamber (#80806; Ibidi). At DIV3, microglia were fixed in 4% PFA/sucrose for 20 min. Next, the cells were incubated with a blocking solution (2% FCS/FBS, 2% BSA, 0.2% fish gelatin, 0.1% Triton X-100) for 30 min. A rat anti-CD68 (1:500; Bio-Rad) was added to blocking solution and incubated overnight at 4°C. The cells were washed three times with PBS and then exposed to a goat anti-rat 555 secondary antibody (1:400) and Hoechst (at 1:2,000) for 45 min. After washing, lipid droplets were stained with BODIPY 493/503 dye (1:1,000; Thermo Fisher Scientific) for 15 min and washed, and then,

200 μl of mounting medium per well was added (50001-4; Ibidi). Images were recorded on a confocal microscope (Leica TCS SP5). A tile scan of a randomly chosen region in a single well of the Ibidi chamber was recorded using a 20x objective at a resolution of 1,024 × 1,024 pixels. These tiles depicted up to 20 microglia. The BODIPY signal was thresholded using Fiji, and BODIPY-positive cells were manually counted. Microglia were isolated from 5 WT and 5 iRhom2$^{-/-}$ animals. Tile scans from different wells were treated as biological replicates, with at least three biological replicates recorded per animal and genotype. Representative images were recorded on a 63x oil immersion objective.

## Statistics and data analysis

All data are presented as the mean ± SD and were analyzed using GraphPad Prism 10.1.2. A two-sided independent $t$ test was performed to compare the means between different genotypes. A one-way or two-way ANOVA with a post hoc test for multiple comparisons was employed, as detailed in the figure legends, for comparing means among three or more groups.

# Data Availability

The mass spectrometry proteomics data have been deposited to the ProteomeXchange Consortium via the PRIDE (Perez-Riverol et al, 2022) partner repository with the dataset identifiers PXD054894 (Secretomics of ADAM17 and iRhom2 KO BV2 cells using high-performance secretome protein enrichment with click sugars) and PXD054898 (Proteomics of ADAM17 and iRhom2 KO microglia using high-performance secretome protein enrichment with click sugars).

# Supplementary Information

# Acknowledgements

We thank Katrin Moschke for excellent technical help. We are grateful to Mathias Jucker (Hertie Institute for Clinical Brain Research, University of Tübingen) that provided the APPPS1 mouse model. This work was funded by the Deutsche Forschungsgemeinschaft (DFG, German Research Foundation) under Germany's Excellence Strategy within the framework of the Munich Cluster for Systems Neurology (EXC 2145 SyNergy—ID 390857198), by the Alzheimer Forschung Initiative e.V., and by the Centers of Excellence in Neurodegeneration through grant CoEN6005. G Jocher was supported by a PhD scholarship from the Friedrich–Naumann–Stiftung für die Freiheit (FNF) with funds from the Bundesministerium für Bildung und Forschung (BMBF). CP Blobel was supported by NIH R35 GM134907.

## Author Contributions

G Jocher: conceptualization, formal analysis, investigation, and writing—original draft, review, and editing.

G Ozcelik: formal analysis, investigation, methodology, and writing—review and editing.

SA Müller: investigation and writing—review and editing.

H-E Hsia: investigation, methodology, and writing—review and editing.

M Lastra Osua: investigation, methodology, and writing—review and editing.

LI Hofmann: investigation and writing—review and editing.

M Aßfalg: investigation and writing—review and editing.

L Dinkel: validation, investigation, and writing—review and editing.

X Feng: investigation and writing—review and editing.

K Schlepckow: resources, validation, and writing—review and editing.

M Willem: resources, validation, and writing—review and editing.

C Haass: resources, validation, and writing—review and editing.

S Tahirovic: formal analysis, validation, investigation, and writing—review and editing.

CP Blobel: resources, validation, and writing—review and editing.

SF Lichtenthaler: conceptualization, supervision, funding acquisition, project administration, and writing—original draft, review, and editing.

## Conflict of Interest Statement

C Haass and K Schlepckow collaborate with Denali Therapeutics, and C Haass is a member of the advisory boards of AviadoBio and Cure Ventures. CP Blobel and the Hospital for Special Surgery have identified iRhom2 inhibitors and have co-founded the start-up company SciRhom in Munich to commercialize iRhom2 inhibitors. H-E Hsia is an employee of Roche Diagnostics GmbH at the time when the article was submitted.

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
