## [Reviewer comments · Life Science Alliance]

Life Science Alliance

The late-onset Alzheimer's disease risk factor RHBDF2 is a modifier of microglial TREM2 proteolysis

Georg Jocher, Gözde Özcelik, Stephan Müller, Hung-En Hsia, Miranda Lastra-Osua, Laura Hofmann, Marlene Aßfalg, Lina Dinkel, Xiao Feng, Kai Schlepckow, Michael Willem, Christian Haass, Sabina Tahirovic, Carl Blobel, and Stefan Lichtenthaler
DOI: <https://doi.org/10.26508/lsa.202403080>

Corresponding author(s): Stefan Lichtenthaler, German Center for Neurodegenerative Diseases and Stefan Lichtenthaler, German Center for Neurodegenerative Diseases

Review Timeline:

Submission Date:	2024-10-09
Editorial Decision:	2024-12-02
Revision Received:	2025-02-21
Editorial Decision:	2025-02-25
Revision Received:	2025-02-26
Accepted:	2025-02-27

Transaction Report:

December 2, 2024

Re: Life Science Alliance manuscript #LSA-2024-03080-T

Prof. Stefan F. Lichtenthaler
DZNE and Technical University Munich
Neuroproteomics
Feodor-Lynen-Str. 17
Munich 81377
Germany

Dear Dr. Lichtenthaler,

Thank you for submitting your manuscript entitled "The late onset Alzheimer's disease risk factor iRhomb2 is a modifier of microglial TREM2 proteolysis" to Life Science Alliance. The manuscript was assessed by expert reviewers, whose comments are appended to this letter. We invite you to submit a revised manuscript addressing the Reviewer comments.

Thank you for this interesting contribution to Life Science Alliance. We are looking forward to receiving your revised manuscript.

Sincerely,

B. MANUSCRIPT ORGANIZATION AND FORMATTING:

Reviewer #1 (Comments to the Authors (Required)):

This is a compelling study that provides interesting insights about the biology of TREM2 in myeloid cells, principally microglia. The authors report iRhom2, an inactive proteinase, as a modifier of ADAM17-mediated TREM2 proteolysis. The authors propose that inhibition/silencing of iRhom2 is protective in an AD pathological context due to inhibition of ADAM17-mediated cleavage of TREM2 resulting in increased TREM2 membrane levels and signaling. Although this is an exciting study, there are some issues that need to be addressed.

A) Figure 1:

Fig. 1A needs to be improved. A more detailed figure /graphical abstract would provide some clarity to the proposed players and their putative interactions. The mechanisms are complex, and the authors need to better place their findings in context. As a general comment, this study would benefit from doing some basic numeric housekeeping which might provide mechanistic insight. For example, in Figures 1, 2, 3 and 4 - The authors should quantify the western blots of 2E and 3D. Also, why did the authors use different loading controls for WB (1B and 1D - Calnexin and B-actin, 2E - Calnexin; 3D- Calnexin and B-actin, 4A- B-actin)? Are these different types of extracts? If so, that should be noted and described in the methods. The level of the mRNAs of TREM2 splice variants should be quantified to control for their contribution to sTREM2 pool. Alternatively, the authors could validate their ELISA and WB findings using an antibody that only detects the cleaved fragment of TREM2 receptor.

B) Figure 2 and 3:

Figure 2D the reduction in sTREM2 is about 75% in iR2 KO, meaning that 25% of sTREM2 must be originating from different pathways. The authors discuss that this factor could be due to the activity of other proteases. However, the authors should highlight that it could also be due to alternative splice isoforms of TREM2 that are secreted without cleavage (PMID: 36805764), which according to some estimates may reach 25% of secreted pool sTREM2 (PMID: 31068200). The TREM2 antibodies used for ELISA target the N-terminal and cannot distinguish between the TREM2 isoforms (related to point previous point in A). Figure 2D and F - The reduction in sTREM2 is about 75% in iR2 KO, however the increase in TREM2 surface levels is about 25-50%. One would expect that the magnitude of sTREM2 reduction in media matched to the increase of surface TREM2, but it is not the case. Could the authors explain this?

Figure 2E and 3D - The mature form of TREM2 is the form that is subjected to cleavage, however both the mature and immature form of TREM2 increase in A17 and iR2 KOs in BV2. The same thing seems to happen in primary microglia but to a lesser extent. Is there an explanation for this? If ADAM17 and iRhom2 only affect the shedding of TREM2, would it be expected that the immature form also changes? Could the authors confirm if the total expression of TREM2 is altered by iRhom2 or ADAM17 KO? This could be cleared by quantifying these immature and mature species of TREM2 as suggested in the previous point A.

C) Figure 4:

Figure 4 - Syk activation is not exclusive to TREM2 signaling. Although iRhom2 KO increases TREM2 membrane levels, it is likely also inducing other changes in the cells (as the authors show, the levels of several peptides change in the media). Having p-Syk increased is suggestive that TREM2 signaling is increased, but not definitive. The authors should re-word the interpretation of this experiment.

D) Figure 5:

Figure 5A - The authors observe an increase in DAM marker genes in iRhom2 KO mice. However, these studies were done in non-pathological mice and the modulation of DAM markers in the absence of a clear pathology may be difficult to interpret. The ideal way to evaluate the DAM phenotype as a response to disease would be to have an amyloid model WT and iRhom2 KO to compare with the respective B6 controls. It is likely not feasible to create this model to address this point, thus, an in vitro approach using a pathological trigger may be important to validate these conclusions.

Figure 5B - Although I appreciate the authors' efforts to use an ex-vivo model that could better mimic microglia biology compared to in vitro, this method does not seem ideal because there are considerable confounding factors. For example, do microglia from

different genotypes have the same survival or migration abilities? Are the stained microglia from the APPS1 mice or from the experimental microglia? These factors could significantly affect the results and interpretations in this model.

E) Discussion:

The discussion is illuminating on many points but is overly long and rambles a bit.

Reviewer #2 (Comments to the Authors (Required)):

The authors provide a detailed and convincing dataset to argue that iRhom2 alters TREM2 proteolysis. They generate genetic knockdown cell line and animal models to enable characterisation of iRhom2 deficiency in multiple microglia and macrophage models providing cross model support for the findings.

I have no concerns about the quality of the experiments, they have been well designed and executed.

I have some minor queries outlined below:

1. With regards to redundancy, sTREM2 is not fully abolished upon iRhom2 KD. Have the authors checked to see if iRhom1 is upregulated in microglia upon iRhom2 KD?
2. Further to point 1, do the authors think modulation of microglial state may alter the residual sTREM2 release observed to attenuate the identified deficits?
3. Could the authors confirm that the background mouse strain for the generation of the A17 and iR2 KO animals are compatible with the C57BL/6J-Tg(Thy1- APPS_w,Thy1-PSEN1*L166P)21Jckr/J) strain? My concern being immunogenic artefacts.
4. Have the authors checked for alterations in microglial secretome due to exposure to N-azido-mannosamine?

Point-by-point response letter

We thank both referees for their positive and constructive reviews. A point-by-point response to all comments is provided below.

Reviewer #1 (Comments to the Authors (Required)):

This is a compelling study that provides interesting insights about the biology of TREM2 in myeloid cells, principally microglia. The authors report iRhom2, an inactive proteinase, as a modifier of ADAM17-mediated TREM2 proteolysis. The authors propose that inhibition/silencing of iRhom2 is protective in an AD pathological context due to inhibition of ADAM17-mediated cleavage of TREM2 resulting in increased TREM2 membrane levels and signaling. Although this is an exciting study, there are some issues that need to be addressed.

A) Figure 1:

Fig. 1A needs to be improved. A more detailed figure /graphical abstract would provide some clarity to the proposed players and their putative interactions. The mechanisms are complex, and the authors need to better place their findings in context. As a general comment, this study would benefit from doing some basic numeric housekeeping which might provide mechanistic insight. For example, in Figures 1, 2, 3 and 4 - The authors should quantify the western blots of 2E and 3D. Also, why did the authors used different loading controls for WB (1B and 1D - Calnexin and B-actin, 2E - Calnexin; 3D- Calnexin and B-actin, 4A- B-actin)? Are these different types of extracts? If so, that should be noted and described in the methods. The level of the mRNAs of TREM2 splice variants should be quantified to control for their contribution to sTREM2 pool. Alternatively, the authors could validate their ELISA and WB findings using an antibody that only detects the cleaved fragment of TREM2 receptor.

This reviewer comment contains 4 points, which we all addressed as follows:

1. As suggested, the scheme in Figure 1A is now updated. Instead of a generic membrane protein substrate, we specifically included TREM2 as a substrate to fit with the manuscript. We also included the TREM2-binding partner DAP12 and the kinase SYK, involved in downstream TREM2 signaling. This addition provides the basis to better understand the experiments in Figure 4. Moreover, an additional scheme is added as supplementary figure S1 and shows in detail how the interaction between ADAM17 and iRhom2 control trafficking, propeptide removal of ADAM17 and, thus, ADAM17 activation.
2. As the reviewer suggested, the TREM2 western blots of 2E and 3D are quantified and added as Supplementary Figure S3.
3. Regarding the comment about the different loading controls: whenever possible we tried to re-probe the original blot with antibodies for the loading controls. When probing the membrane with iRhom2, ADAM17, or TREM2 antibodies, this required, however, choosing loading controls that do not interfere with the signal of the previously used antibody, such as iRhom2 or ADAM17. The selection of loading controls also considered the species origin of the primary antibodies to ensure

compatibility and avoid cross-reactivity. As a result, the choice of the loading control was individually based on the choice of the initially used antibodies.

4. As a last point, the reviewer suggested quantifying alternative sTREM2 splice variants, which may not arise through proteolytic cleavage. This comment is based on our result that the knock-out of ADAM17 or iRhom2 in Fig. 2A did largely, but not fully block sTREM2 release, with about 10-25% sTREM2 remaining in the supernatants of the knock-out cells. We had speculated that the remaining sTREM2 may arise through cleavage by another protease, in particular ADAM10, a close homolog of ADAM17. We have now chosen a pharmacological approach to show that this is indeed the case. We treated the BV2 cells in Fig. 2 with 10 μ M GI254023X (GI), an ADAM10-preferring inhibitor, or with BB94, a broader-spectrum metalloproteinase inhibitor, which blocks both ADAM10 and ADAM17 as well as a few related proteases. The results are now included in new Supplementary Figure S2. Our findings show that ADAM10 inhibition partly reduces sTREM2 release in control cells (NTC) and further reduces the low-levels of remaining sTREM2 in the ADAM17- and iRhom2-knock-out cells. The broader-spectrum inhibitor BB94 completely inhibited the remaining sTREM2 release from the knock-out cells. Thus, we conclude that sTREM2 in this cellular BV2 system is released by proteolytic cleavage, mostly by ADAM17 and, to a lower extent, by ADAM10 and potentially related proteases. Our results do not rule out the possibility of alternative sTREM2 splice variants, but quantitatively they appear to have a minor role in sTREM2 protein levels in the BV2 cell supernatants.

We added the following paragraph to the results section on page 5: “The reduction of sTREM2 in iRhom2^{-/-} and ADAM17^{-/-} BV2 cells, as quantified by mass spectrometry, was further validated by an ELISA assay in the same conditioned media, where reductions of ~90% and ~75% of sTREM2 were measured, respectively, upon loss of ADAM17 and iRhom2 (Fig. 2D). The remaining sTREM2 in the knock-out cell supernatants may either be proteolytically generated by other proteases, such as ADAM10, or represent soluble, secreted forms of TREM2 that result from alternative splicing of the TREM2 mRNA (Del-Aguila et al, 2019; Moutinho et al, 2023). To distinguish between both possibilities, we treated the iRhom2^{-/-} and ADAM17^{-/-} BV2 cells with the ADAM10-preferring inhibitor GI254023X or the broader-spectrum metalloprotease inhibitor BB94, which blocks ADAM10 and ADAM17 as well as a few additional metalloproteases. Treatment of the knock-out cells with either GI254023X or BB94 blocked sTREM2 nearly completely (Suppl. Fig. S2). We conclude that sTREM2 in the BV2 cells is predominantly generated through proteolytic cleavage, mostly by ADAM17 and to a lower extent by ADAM10, rather than by alternative mechanisms, such as the generation of alternatively spliced, secreted sTREM2 variants.”

B) Figure 2 and 3:

Figure 2D the reduction in sTREM2 is about 75% in iR2 KO, meaning that 25% of sTREM2 must be originating from different pathways. The authors discuss that this factor could be due to the activity of other proteases. However, the authors should highlight that it could also be due to alternative splice isoforms of TREM2 that are secreted without cleavage

(PMID: 36805764), which according to some estimates may reach 25% of secreted pool sTREM2 (PMID: 31068200). The TREM2 antibodies used for ELISA target the N-terminal and cannot distinguish between the TREM2 isoforms (related to point previous point in A). Figure 2D and F - The reduction in sTREM2 is about 75% in iR2 KO, however the increase in TREM2 surface levels is about 25-50%. One would expect that the magnitude of sTREM2 reduction in media matched to the increase of surface TREM2, but it is not the case. Could the authors explain this?

We addressed both reviewer points in this comment in the following way:

1. As suggested, we added the mentioned references and now clarify that sTREM2 may result not only from proteolytic cleavage of full-length TREM2, but also from the expression of sTREM2 splice variants. Given the results of the pharmacological experiments carried out to address this reviewer's comment A above, we added new suppl. Fig. 2 and describe in the results section that in our BV2 cell system, the detected sTREM2 is released proteolytically. As mentioned in our answer to comment A above, the new text in the results section on page 5 is: "The reduction of sTREM2 in iRhom2^{-/-} and ADAM17^{-/-} BV2 cells, as quantified by mass spectrometry, was further validated by an ELISA assay in the same conditioned media, where reductions of ~90% and ~75% of sTREM2 were measured, respectively, upon loss of ADAM17 and iRhom2 (Fig. 2D). The remaining sTREM2 in the knock-out cell supernatants may either be proteolytically generated by other proteases, such as ADAM10, or represent soluble, secreted forms of TREM2 that result from alternative splicing of the TREM2 mRNA (Del-Aguila *et al*, 2019; Moutinho *et al*, 2023). To distinguish between both possibilities, we treated the iRhom2^{-/-} and ADAM17^{-/-} BV2 cells with the ADAM10-preferring inhibitor GI254023X or the broader-spectrum metalloprotease inhibitor BB94, which blocks ADAM10 and ADAM17 as well as a few additional metalloproteases. Treatment of the knock-out cells with either GI254023X or BB94 blocked sTREM2 nearly completely (Suppl. Fig. S2). We conclude that sTREM2 in the BV2 cells is predominantly generated through proteolytic cleavage, mostly by ADAM17 and to a lower extent by ADAM10, rather than by alternative mechanisms, such as the generation of alternatively spliced, secreted sTREM2 variants."
2. This reviewer comment also refers to the observation that the reduction of sTREM2 release (75%) is stronger than the corresponding increase at the cell surface (25%). We agree that one may expect both values to be similar. But this assumption holds only true if all TREM2 molecules at the surface undergo shedding and if the shedding exclusively takes place at the plasma membrane. If, however, only a fraction of the surface TREM2 undergoes shedding during the experimental time, a strong reduction of sTREM2 may only lead to a small increase in total surface TREM2. Likewise, if a fraction of TREM2 is shed within the secretory pathway (i.e. the TGN) before reaching the plasma membrane, a reduced shedding may not alter surface TREM2 levels, while still increasing total mature TREM2 in the lysate. In fact, while it is often assumed that surface proteins get cleaved at the surface, this has rarely been shown for the hundreds of membrane proteins known to undergo shedding (Lichtenthaler *et al*. EMBO J 2018). For example, for the AD-linked APP protein we still do not know where its shedding by ADAM proteases takes place – either at the plasma membrane or in the TGN or at both cellular locations. As we do not know what percentage of

TREM2 molecules undergoes shedding, we have not included the above considerations into the manuscript.

Figure 2E and 3D - The mature form of TREM2 is the form that is subjected to cleavage, however both the mature and immature form of TREM2 increase in A17 and iR2 KOs in BV2. The same thing seems to happen in primary microglia but to a lesser extent. Is there an explanation for this? If ADAM17 and iRhom2 only affect the shedding of TREM2, would it be expected that the immature form also changes? Could the authors confirm if the total expression of TREM2 is altered by iRhom2 or ADAM17 KO? This could be cleared by quantifying these immature and mature species of TREM2 as suggested in the previous point A.

As suggested, we quantified mature and immature TREM2 in Fig. 2D (BV2 cells) and Fig. 3D (primary microglia). The quantification is now included as new Supplementary Figure S3. Mature TREM2 was increased in both cell types upon knock-out of ADAM17 or iRhom2, as expected if there is less proteolytic cleavage of TREM2. In the primary microglia, immature TREM2 was not significantly increased upon iRhom2 knock-out, consistent with the lack of increased TREM2 expression in primary microglia, as seen in the nanostring RNA analysis in Fig. 5A (where TREM2 is now labeled). In the BV2 cells, loss of ADAM17 and – to a lesser extent – loss of iRhom2, also increased immature TREM2. We cannot fully explain why loss of TREM2 shedding in BV2 cells would also increase immature TREM2, but we can offer a speculation. TREM2 is a cell surface protein that traffics through the secretory pathway to the plasma membrane. If there is a strong increase in mature TREM2, as seen upon inhibition of TREM2 shedding, a sort of traffic jam appears possible which prevents efficient maturation of further immature TREM2 molecules, leading effectively to an increase of immature TREM2. Given that this remains a speculation, we did not include this speculation into the manuscript. In any case, loss of ADAM10 and iRhom2 in the microglia efficiently reduced TREM2 shedding, even if there is somewhat more total TREM2 in the BV2 cells. Yet, we did include the information that TREM2 expression was not altered in the iRhom2^{-/-} microglia (Results, page 6): “Yet, other genes associated with the DAM state, such as TREM2, did not show altered expression in iRhom2-deficient microglia (Fig. 5A).”

C) Figure 4:

Figure 4 - Syk activation is not exclusive to TREM2 signaling. Although iRhom2 KO increases TREM2 membrane levels, it is likely also inducing other changes in the cells (as the authors show, the levels of several peptides change in the media). Having p-Syk increased is suggestive that TREM2 signaling is increased, but not definitive. The authors should re-word the interpretation of this experiment.

As suggested, we adjusted the wording. The paragraph in the results' section on page 6 is now: “Phosphorylation of the kinase SYK (pSyk) is a commonly observed downstream event in the signaling of TREM2 and additional cell surface receptors (Schlepckow et al, 2020; van Lengerich et al, 2023; Wang et al., 2022).

As expected, immunoblot analysis of iRhom2^{-/-} BMDMs showed a lack of iRhom2 expression and impaired ADAM17 maturation (Fig. 4A), consistent with the findings in BV2 cells and primary microglia (Fig. 1B, D). Notably, pSyk levels were doubled in iRhom2-deficient BMDMs compared to wild-type controls (Fig. 4A, B), consistent with the notion that the loss

of iRhom2 may enhance TREM2 signaling. Additionally, we observed a more than 50% reduction in soluble TREM2 production, mirroring the results seen in BV2 cells and primary microglia (Fig. 1 C, E).

These findings lead us to conclude that iRhom2 controls downstream SYK signaling.”

D) Figure 5:

Figure 5A - The authors observe an increase in DAM marker genes in iRhom2 KO mice. However, these studies were done in non-pathological mice and the modulation of DAM markers in the absence of a clear pathology may be difficult to interpret. The ideal way to evaluate the DAM phenotype as a response to disease would be to have an amyloid model WT and iRhom2 KO to compare with the respective B6 controls. It is likely not feasible to create this model to address this point, thus, an in vitro approach using a pathological trigger may be important to validate these conclusions.

Indeed, the DAM phenotype of microglia typically requires a pathological stimulus, such as amyloid pathology in mice. The DAM state is characterized by increased expression of several genes, including *Clec7A*, *Itgax* and *TREM2*, as initially shown by Keren-Shaul et al. (Cell 2017). The iRhom2 KO microglia analyzed in our study were not from an AD mouse model and only showed mildly increased RNA levels of a few genes associated with the DAM phenotype, such as *Clec7a* and *Itgax*. Yet, other genes associated with the DAM phenotype, e.g. *TREM2*, did not show an upregulated expression (see Fig. 5A, *TREM2* is now labeled in the volcano plot). We conclude that iRhom2 KO microglia do not show the full DAM phenotype, but share some gene expression changes that are also seen in DAM microglia, even though with a smaller effect size. We tried to describe this phenotype by calling it DAM-like, e.g. in the results section in the following sentence: “This suggests that iRhom2^{-/-} microglia are in a state with DAM-like properties.” The reviewer comment shows us that our expression “DAM-like” may have sounded overstated. Thus, we toned down the description of the phenotype. The corresponding paragraph in the results section on page 6 is now: “Gene expression of acutely isolated microglia was analyzed using a Nanostring panel that included 235 selected genes related to microglia activation, revealing an increase larger than 35% in expression of DAM-linked genes *Clec7a*, *Itgax*, *Cst7*, and *Lpl* (Fig. 5A) in iRhom2-deficient microglia. The increased expression of these genes is a hallmark of DAM, whereas genes indicative of a homeostatic signature remained unaltered (*Csf1r*, *Tgfb1*, *P2ry12*). Yet, other genes associated with the DAM state, such as *TREM2*, did not show altered expression in iRhom2-deficient microglia (Fig. 5A). These findings suggest that iRhom2^{-/-} microglia are not in a DAM state but in a state which shares some gene expression change with the DAM state.” We also deleted the expression DAM in the last sentence of the results section and the second paragraph of the discussion.

We agree with the reviewer that a pathological challenge, e.g. with Abeta pathology in vitro or in vivo, may allow to determine whether and how strongly a loss of iRhom2 alters the DAM state of microglia. We have not carried out the suggested in vitro stimulation experiment for the following two reasons. First, our study investigates the physiological and not the pathophysiological function of iRhom2. Second, we used proteomics and tested in a previous, unpublished experiment, how aggregated Abeta modulates the activity state of primary murine wild-type microglia. As a control, we used LPS, a strong activator of microglia, which, however, does not induce a DAM-phenotype, but leads to a different activation state of microglia. Compared to LPS, the Abeta stimulus changed protein expression of only very few genes and did not induce a DAM phenotype in our experiment. Thus, we feel that the suggested Abeta stimulation in vitro may not be best suited to answer

whether amyloid pathology alters the activity state of iRhom2-deficient microglia. Instead, we plan to mate iRhom2-deficient mice with an AD mouse model, which, however, is not feasible quickly enough for the revision process, as also noted by the reviewer.

Figure 5B - Although I appreciate the authors' efforts to use an ex-vivo model that could better mimic microglia biology compared to in vitro, this method does not seem ideal because there are considerable confounding factors. For example, do microglia from different genotypes have the same survival or migration abilities? Are the stained microglia from the APPS1 mice or from the experimental microglia? These factors could significantly affect the results and interpretations in this model.

We agree with the reviewer that the raised points could affect the results. For that reason, the method is set up in such a way that we avoid these potentially confounding factors. First, we normalized the microglia-cleared plaque area by the number of microglia, using the number of CD68-positive particles counted on the sections. In this way, we exclude effects of survival, proliferation or migration interfering with the calculation of the plaque clearance efficiency shown in Fig. 5B. Second, to ensure that we only stain the microglia added to the brain slices, but not the microglia within the brain slice, we permeabilized the tissue section for only 3 min, which allowed us to permeabilize and stain only the exogenously added microglia at the surface of the slice, but not the microglia within the slice.

We added the following sentence to the results section on page 7 :” This cleared area is normalized for the number of exogenously added microglia on the brain section, as measured by CD68 staining.” We also added a few sentences to the corresponding methods description on page 14: “The cleared Ab area (ThR staining) was then normalized for the number of exogenously added microglia on the brain section, as measured by CD68 staining. To ensure that only the exogenously added microglia, but not the microglia within the brain section, were used for CD68 staining, the fixed slices were permeabilized for only 3 minutes, which permeabilizes only the exogenously added microglia at the surface of the slice. “

E) Discussion:

The discussion is illuminating on many points but is overly long and rambles a bit. As suggested, we substantially shortened the discussion section.

Reviewer #2 (Comments to the Authors (Required)):

The authors provide a detailed and convincing dataset to argue that iRhom2 alters TREM2 proteolysis. They generate genetic knockdown cell line and animal models to enable characterisation of iRhom2 deficiency in multiple microglia and macrophage models providing cross model support for the findings.

I have no concerns about the quality of the experiments, they have been well designed and executed.

I have some minor queries outlined below:

1. With regards to redundancy, sTREM2 is not fully abolished upon iRhom2 KD. Have the authors checked to see if iRhom1 is upregulated in microglia upon iRhom2 KD?

This comment is based on our result that the knock-out of ADAM17 or iRhom2 in Fig. 2D (BV2 cells) and Fig. 3C (microglia) did largely, but not fully block sTREM2 release, with about 10-25% sTREM2 remaining in the supernatants of the knock-out cells. A similar point had been raised by reviewer 1, who suggested that the remaining sTREM2 may represent soluble, secreted TREM2 splice variants that do not even require proteolytic processing. We had speculated that the remaining sTREM2 may arise through cleavage by another protease, in particular ADAM10, a close homolog of ADAM17. We have now chosen a pharmacological approach to show that this is indeed the case. We treated the BV2 cells in Fig. 2 with GI254023X (GI), an ADAM10-preferring inhibitor, or with BB94, a broader-spectrum metalloproteinase inhibitor, which blocks both ADAM10 and ADAM17 as well as a few related proteases. The results are now included in new Supplementary Figure S2. Our findings show that ADAM10 inhibition partly reduced sTREM2 release in control cells (NTC) and further reduced the low levels of remaining sTREM2 in the ADAM17- and iRhom2-knock-out cells. The broader-spectrum inhibitor BB94 completely inhibited the remaining sTREM2 release from the knock-out cells. Thus, we conclude that sTREM2 in this cellular BV2 system is released by proteolytic cleavage, mostly by ADAM17 and to lower extent by ADAM10 and potentially related proteases. Our results do not rule out the possibility of alternative sTREM2 splice variants, but quantitatively they appear to have a minor role in sTREM2 protein levels in the BV2 cell supernatants. Additionally, as iRhom1 and iRhom2 are specific subunits of ADAM17, but not of ADAM10 or other metalloproteases, we do not need to imply an involvement of iRhom1 in the remaining sTREM2 levels upon knock-out of iRhom2 or ADAM17.

We added the following paragraph to the results section on page 5: *“The reduction of sTREM2 in iRhom2^{-/-} and ADAM17^{-/-} BV2 cells, as quantified by mass spectrometry, was further validated by an ELISA assay in the same conditioned media, where reductions of ~90% and ~75% of sTREM2 were measured, respectively, upon loss of ADAM17 and iRhom2 (Fig. 2D). The remaining sTREM2 in the knock-out cell supernatants may either be proteolytically generated by other proteases, such as ADAM10, or represent soluble, secreted forms of TREM2 that result from alternative splicing of the TREM2 mRNA (Del-Aguila et al, 2019; Moutinho et al, 2023). To distinguish between both possibilities, we treated the iRhom2^{-/-} and ADAM17^{-/-} BV2 cells with the ADAM10-preferring inhibitor GI254023X or the broader-spectrum metalloprotease inhibitor BB94, which blocks ADAM10 and ADAM17 as well as a few additional metalloproteases. Treatment of the knock-out cells with either GI254023X or BB94 blocked sTREM2 nearly completely (Suppl. Fig. S2). We conclude that sTREM2 in the BV2 cells is predominantly generated through proteolytic cleavage, mostly by*

ADAM17 and to a lower extent by ADAM10, rather than by alternative mechanisms, such as the generation of alternatively spliced, secreted sTREM2 variants.”

2. Further to point 1, do the authors think modulation of microglial state may alter the residual sTREM2 release observed to attenuate the identified deficits?

As discussed under point 1 raised by this reviewer, we now demonstrate that the remaining sTREM2 in iRhom2-deficient cells stems from metalloproteases other than ADAM17, in particular ADAM10.

Altered microglial cell states, such as DAM, can indeed lead to altered expression of TREM2, both at the RNA (e.g. Keren-Shaul et al. Cell 2017) and protein level (e.g. Sebastian Monasor et al. eLife 2020). With more TREM2 protein being available, it is well possible and even likely that more TREM2 will be shed, resulting in higher sTREM2 in the medium.

Yet, our nanostring analysis did not reveal increased TREM2 RNA in iRhom2-deficient microglia (Fig. 5A). Additionally, only a few DAM-related genes were mildly upregulated upon iRhom2-deficiency, indicating that the iRhom2-deficient microglia are not in a full DAM state. Thus, it appears unlikely that lack of iRhom2, besides reducing ADAM17-mediated TREM2 cleavage, additionally mildly increases TREM2 shedding due to altered TREM2 expression.

3. Could the authors confirm that the background mouse strain for the generation of the A17 and iR2 KO animals are compatible with the C57BL/6J-Tg(Thy1- APPSw,Thy1- PSEN1*L166P)21Jckr/J)) strain? My concern being immunogenic artefacts.

This comment refers to the plaque-clearing assay in Fig. 5B. The iRhom2 KO animals were generated on a mixed BL6/129 background and backcrossed 9x to the C57BL/6 background (McIlwain et al. Science 2012). Importantly, for the isolation of wild-type and iRhom2 KO microglia, littermate control mice were used to ensure that the observed differences in plaque-clearing efficiency were due to the loss of iRhom2 and not due to strain differences. As an additional information on the assay, which has been used for several studies: The plaque-clearing efficiency that we observe for mouse microglia is even similar to microglia from other species and origins, such as iPS-derived human monocytes (Claes et al. AlzheimersDement 2019) and human macrophages (Colombo et al. Nat Comm 2021).

4. Have the authors checked for alterations in microglial secretome due to exposure to N-azido-mannosamine?

We have not specifically tested this possibility for the microglia in this study, but have tested it previously for several other cell types, including HEK293 cells, and primary murine neurons. One specific example of our tests was to show that N-azido-mannosamine labeling did not alter APP shedding or its processing by beta-secretase in primary murine neurons (Kuhn et al. EMBO J 2012). Importantly, N-azido-mannosamine is converted in cells into N-azido-sialic acid and incorporated into glycan structures of proteins as one of the terminal sugars, which is less likely to interfere with protein function/secretion compared to incorporating the azido moiety as a modified amino acid (AHA) directly into the protein backbone (Eichelbaum et al. Nat Biotech 2012). In fact, while AHA labeling can lead to cell toxicity, we have never observed toxicity with N-azido-mannosamine when testing numerous different cell types, including primary murine neurons, astrocytes, microglia and oligodendrocytes (Tüshaus et al. EMBO J 2020). Lastly, in our experiments in this manuscript, both the wild-type and the knock-out cells were labeled in exactly the same way with

ManNAz. Thus, even if there were subtle changes to the secretome based on the addition of ManNAz, such changes would be seen in both wild-type and knock-out conditions. Thus, we are confident that the changes that we observed were due to the knock-out and not due to ManNAz addition.

Additional change to the manuscript

One more change was made to the manuscript that was not specifically requested by the reviewers:

In Fig. 1B we replaced the ADAM17 blot with a new blot from the same samples, but where the lysates had been deglycosylated with PNGaseF before Western blotting. It is known that the pro- and mature ADAM17 bands are better resolved on blots after deglycosylation. And this is also seen in Fig. 1D, where we had loaded the deglycosylated samples already in the first version of the manuscript. Thus, the new blot for Fig. 1B is now easier to understand for the readers as the samples were treated in exactly the same way as in Fig. 1D. The new blot gives the same conclusion as the old blot: upon ADAM17 knock-out, both pro- and mature ADAM17 are absent, whereas upon iRhom2 knock-out only the mature ADAM17 is absent.

February 25, 2025

RE: Life Science Alliance Manuscript #LSA-2024-03080-TR

Prof. Stefan F. Lichtenthaler
German Center for Neurodegenerative Diseases
Neuroproteomics
Feodor-Lynen-Str. 17
Munich 81377
Germany

Dear Dr. Lichtenthaler,

Thank you for submitting your revised manuscript entitled "The late-onset Alzheimer's disease risk factor iRhom2 is a modifier of microglial TREM2 proteolysis". We would be happy to publish your paper in Life Science Alliance pending final revisions necessary to meet our formatting guidelines.

- please be sure that the authorship listing and order is correct
- please add the X and Bluesky handles of your host institute/organization as well as your own or/and one of the authors in our system
- please note that the titles in the system and manuscript file must match
- the datasets listed in the Data Availability statement should be made publicly accessible at this point, removing the need for Reviewer access information

A. FINAL FILES:

B. MANUSCRIPT ORGANIZATION AND FORMATTING:

Sincerely,

February 27, 2025

RE: Life Science Alliance Manuscript #LSA-2024-03080-TRR

Prof. Stefan F Lichtenthaler
German Center for Neurodegenerative Diseases
Neuroproteomics
Feodor-Lynen-Str. 17
Munich 81377
Germany

Dear Dr. Lichtenthaler,

Thank you for submitting your Research Article entitled "The late-onset Alzheimer's disease risk factor RHBDF2 is a modifier of microglial TREM2 proteolysis". It is a pleasure to let you know that your manuscript is now accepted for publication in Life Science Alliance. Congratulations on this interesting work.

DISTRIBUTION OF MATERIALS:

Again, congratulations on a very nice paper. I hope you found the review process to be constructive and are pleased with how the manuscript was handled editorially. We look forward to future exciting submissions from your lab.

Sincerely,
